# Nicotine induces abnormal motor coupling through sensitization of a mechanosensory circuit in *Caenorhabditis elegans*

Yuting Liu[1ʘ], Leiru Huang[1ʘ], Ruipeng Wang[1ʘ], Fukang Qi[1], Yiwen Liu[1], Qingyuan Chen[1], Morgane Mialon[2], Lili Chen[1], Berangere Pinan-Lucarre[2], Shangbang Gao[1]*

1 Key Laboratory of Molecular Biophysics of the Ministry of Education, College of Life Science and Technology, Huazhong University of Science and Technology, Wuhan, China, 2 Melis, Universite Claude Bernard Lyon 1, CNRS UMR5284, INSERM U1314, Institut NeuroMyoGene—Faculte de Medecine et de Pharmacie, Lyon, France

ʘ These authors contributed equally to this work.
* sgao@hust.edu.cn

## Abstract

Nicotine exposure elicits diverse behavioral changes, yet the underlying neural pathways and molecular mechanisms remain incompletely understood. Here, we demonstrate that chronic nicotine exposure markedly increases both the initiation and duration of reversals in *Caenorhabditis elegans*. Strikingly, these reversals were tightly coupled with the rhythmic body contractions of the defecation motor program (DMP). Through pharmacological, genetic, in situ electrophysiological, and calcium imaging analyses, we show that nicotine enhances the activity of the AVA interneuron via selective upregulation of ACR-16, a nicotinic ACh receptor critical for nicotine-induced motor coupling. Ablation of touch receptor neurons (TRNs) or inhibition of TRNs-mediated mechanosensation completely abolished this coupling. Furthermore, optogenetic activation of TRNs in nicotine-treated animals evoked stronger AVA depolarization, and nicotine amplified gentle touch-evoked reversals. Together, these findings reveal a potential interoceptive effect of nicotine mediated by sensitization of the TRNs-AVA mechanosensory pathway, providing new insight into the neural and molecular basis of nicotine's modulation of sensory-motor coupling.

## Introduction

Smoking poses severe health risks, and nicotine—the primary psychoactive component of tobacco—exerts potent rewarding effects by upregulating nicotinic acetylcholine receptors (nAChRs) in dopaminergic neurons, particularly within brain regions such as the ventral tegmental area [1–3]. This receptor upregulation drives dopamine release, a key mechanism underlying nicotine addiction [4–6]. During withdrawal, the sudden absence of nicotine reduces dopamine release, contributing to a range of neurological and psychiatric symptoms [7]. Beyond dopamine dysregulation,

**Data availability statement:** All data generated for this study are available in the Supplementary Information files and from the public Zenodo repository (DOI: https://doi.org/10.5281/zenodo.17129063).

**Funding:** This research was supported by the National Natural Science Foundation of China (32371189 to S.G.; 32300984 to L.C., https://www.nsfc.gov.cn/), the Major International (Regional) Joint Research Project (32020103007 to S.G., https://www.nsfc.gov.cn/), the National Key Research and Development Program of China (2022YFA1206000 to S.G., https://www.most.gov.cn/), the Basic Research Support Program of Huazhong University of Science and Technology (2025BRA004 to S.G., http://kfy.hust.edu.cn/), Double First-Class Construction Funds of Hubei Province (5001170159 to S.G. https://kjt.hubei.gov.cn). The funders did NOT play any role in the study design, data collection and analysis, decision to publish, or preparation of the manuscript.

**Competing interests:** The authors have declared that no competing interests exist.

**Abbreviations :** ACh, acetylcholine; ATR, all-trans retinal; ChR2, Channelrhodopsin-2; DMP, defecation motor program; EPSPs, excitatory postsynaptic potentials; nAChRs, nicotinic acetylcholine receptors; NGM, Nematode Growth Medium; TRNs, touch receptor neurons.

additional mechanisms—including elevated corticotropin-releasing factor activity [8] and altered GABAergic and glutamatergic transmission [9,10]—also shape nicotine's effects. These changes underlie withdrawal symptoms such as cravings and mood disturbances, which often lead to relapse and complicate smoking cessation.

Chronic heavy smokers, exposed to sustained high nicotine concentrations, frequently experience more severe withdrawal symptoms [11,12]. Recent evidence indicates that activation of upstream GABAergic neurons plays a crucial role in regulating dopamine release and driving nicotine-dependent behavioral states [13–15]. Given the profound health consequences of smoking and incomplete understanding of the neural mechanisms underlying high-dose nicotine effects, it is essential to further dissect the neural circuits and molecular targets mediating these responses.

The nematode *Caenorhabditis elegans* displays well-defined behavioral responses to nicotine, including acute reactions, adaptation, and withdrawal-like symptoms [16–22]. Similar to mammals, nicotine responses in *C. elegans* are dose-dependent [16,18]. At low concentrations (0.5–5 µM), worms exhibit acute nicotine responses and withdrawal symptoms upon cessation [16,18]. At moderate (0.1 mM) to high (1 mM) concentrations, nicotine alters locomotion speed [18], while at extremely high concentrations (>20 mM), *C. elegans* undergoes behavioral adaptation and eventual paralysis [18,20,22]. Low-dose nicotine responses require nAChRs, which are linked to TRP channels [16], and strikingly, these behaviors can be rescued in mutants by expressing the mammalian α4β2 nAChR, underscoring the evolutionary conservation of nicotine's effects. In contrast, resistance to high-dose nicotine has been linked to genes such as *tax-6*, *nra-1*, and *soc-1* [20]. Prolonged exposure to extremely high nicotine concentrations (30–31 mM) reduces nicotinic receptor abundance and suppresses egg-laying [22]. Despite these advances, the neural circuits and molecular mechanisms underlying high-dose nicotine-induced behavioral modulation remain poorly defined.

In this study, we report that high-concentration nicotine (1 mM) induces marked locomotor alterations in *C. elegans*, including frequent reversals and increased locomotion speed. Unexpectedly, these reversals are strongly coupled to the rhythmic body contractions of the defecation motor program (DMP), suggesting enhanced interoceptive sensitivity to internal mechanical state. High-dose nicotine selectively upregulates the α7-like nAChR ACR-16 [23] in the AVA command interneuron, which mediates the coupling between DMP and reversals. Mechanosensory input from touch receptor neurons (TRNs) is indispensable for this behavior: TRNs ablation or mechanosensation blockade completely abolished the nicotine-induced coupling. Moreover, nicotine enhanced TRN-to-AVA signal strength and prolonged gentle touch-evoked reversals. Together, these findings reveal that nicotine induces abnormal motor coupling via sensitization of a mechanosensory neural circuit in *C. elegans*, offering insight into how nicotine might trigger anxiety-like or aversive behaviors through altered sensory-motor integration.

## Results

### Nicotine increases reversal frequency in *C. elegans*

To investigate the behavioral effects of nicotine, we developed a brief chronic nicotine exposure assay in which young adult hermaphrodites were incubated with 1 mM nicotine for 3 h (Fig 1A; Materials and methods). Following exposure, animals were transferred to nicotine-free Nematode Growth Medium (NGM) agar plates seeded with a thin layer of *Escherichia coli* OP50 for behavioral analysis. Under these conditions, worms exhibited a robust nicotine-dependent behavioral response [24].

Compared with nicotine-naïve controls (− nic), nicotine-exposed worms (+ nic) displayed disrupted forward locomotion, characterized by discontinuous crawling trajectories, markedly increased reversal frequency, and elevated locomotion speed (Fig 1B). The most prominent difference was the substantial increase in reversal frequency and movement velocity. Whereas spontaneous backward movements in control worms were brief and sporadic, nicotine-exposed animals exhibited reversals that were highly rhythmic, significantly prolonged, and executed at faster speeds (S1A Fig). Consequently, the proportion of time spent moving backward was markedly greater in the nicotine-treated group (S1B Fig). Quantitatively, reversal frequency increased nearly 3-fold (− nic, $3.3 \pm 0.34$/3 min, $n = 22$; + nic, $9.3 \pm 0.46$/3 min, $n = 22$, $p < 0.001$) (S1C Fig), reversal duration was extended more than 4-fold (− nic, $1 \pm 0.08$ s, $n = 22$; + nic, $4.7 \pm 0.27$ s, $n = 22$, $p < 0.001$) (S1D Fig), and reversal speed was significantly elevated (− nic, $0.09 \pm 0.003$ mm/s, $n = 22$; + nic, $0.12 \pm 0.003$ mm/s, $n = 22$, $p < 0.001$) (S1E Fig). Meanwhile, forward locomotion speed was also enhanced (− nic, $0.11 \pm 0.003$ mm/s, $n = 22$; + nic, $0.13 \pm 0.004$ mm/s, $n = 22$, $p < 0.001$) (S1F Fig). These high-frequency, prolonged, high-speed reversals resemble nicotine-induced anxious or aversive behaviors in mammals [25,26].

### Nicotine-enhanced reversals are temporally coupled with the defecation motor program

Nicotine-induced behavior changes have been reported across multiple model organisms [26,27]. Here, we observed that nicotine-evoked high-frequency reversals in *C. elegans* exhibited a striking rhythmicity. These reversals occurred approximately every 45–50 s, prompting us to consider whether they might be linked to an intrinsic rhythmic process. We analyzed locomotion over 10 consecutive cycles of the DMP in wild-type animals. In controls, ~20% of DMP cycles were followed by short backward movements (Fig 1C and 1D and S1 Movie), consistent with previous reports of DMP-locomotion coupling [28]. By contrast, nicotine drove nearly 100% of DMP cycles to trigger reversals immediately following DMP onset, indicating robust DMP-reversal coupling (Fig 1C and 1D and S2 Movie). This suggested that nicotine's effect arises from modulation of intrinsic rhythmic signaling, rather than simply producing general excitatory drive.

To distinguish between these possibilities, we examined the effect of glutamate—another neurotransmitter implicated in reward regulation [29] and nicotine dependence [10]—under identical conditions. Glutamate exposure did not alter the proportion of DMP-reversal coupling (Fig 1D). Furthermore, although nicotine affects gastrointestinal motility in mammals [30], the DMP cycle length in *C. elegans* remained unchanged after nicotine exposure (S2A–S2C Fig). These results suggest that nicotine-induced motor coupling does not stem from altered gut rhythmicity or nonspecific excitation.

Interestingly, nicotine-enhanced couplings showed <5 s delays between DMP onset and reversal initiation, in contrast to previous observations in which reversals typically occurred ~10 s before DMP [28]. The DMP consists of three sequential phases: posterior body wall muscle contraction (pBoc), followed by anterior body wall muscle contraction (aBoc), and the final enteric muscle expulsion (Exp) [31] (S2A Fig). To determine whether nicotine alters the coupling temporal relationship between DMP phases and reversal, we categorized reversal onset times relative to these phases and found that nicotine-driven reversals occurred predominantly during pBoc (S2D Fig).

We next assessed coupling in mutants with phase-specific DMP defect. In *egl-8* mutants (defective pBoc), nicotine-enhanced coupling was strongly reduced, whereas *nlp-40* and *aex-2* mutants (impaired Exp) [31,32] showed no significant change (S2E Fig). *aex-3* mutants, defective in both aBoc and Exp [31,33], exhibited partial reduction, but less than *egl-8*.

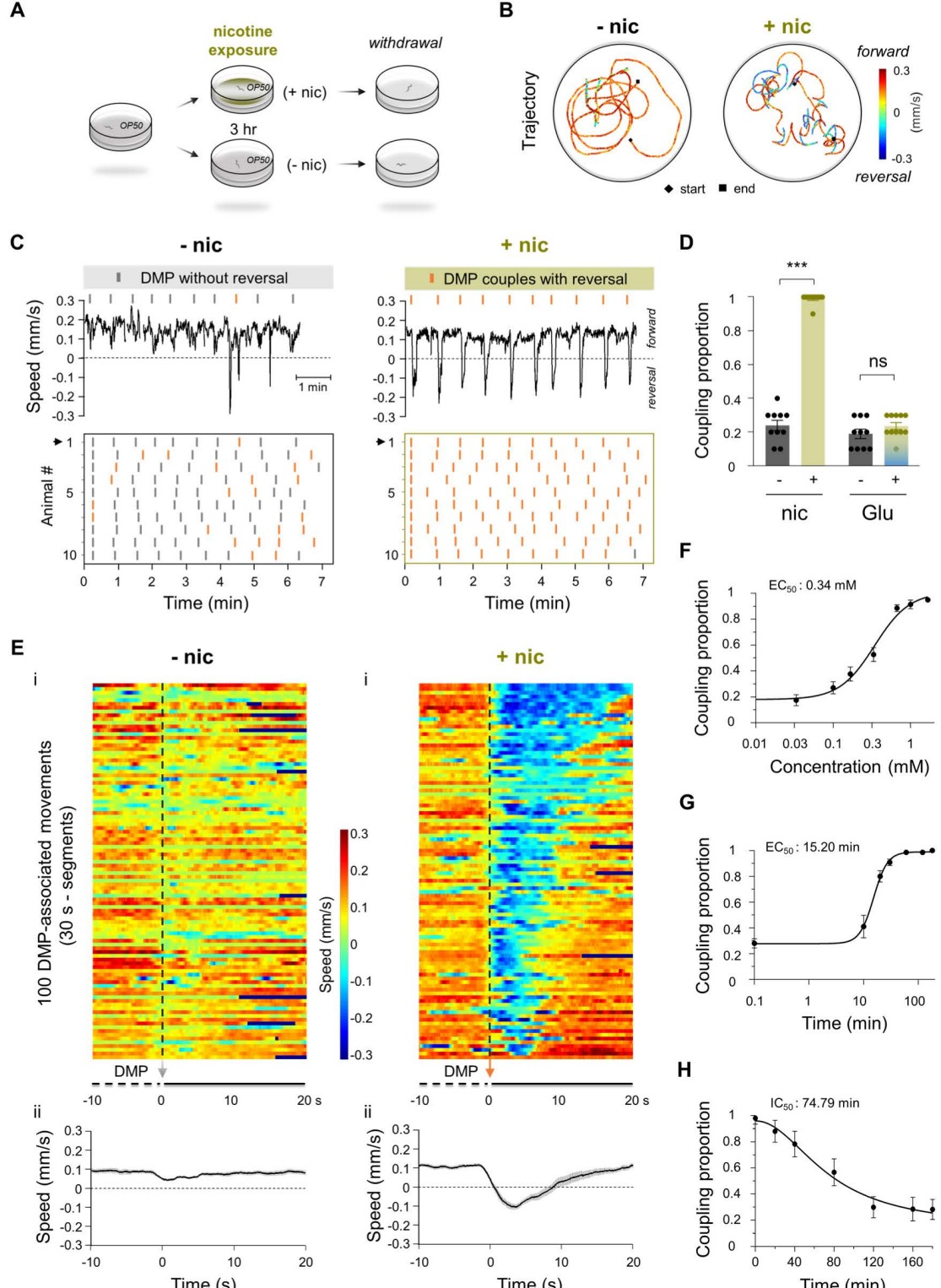

**Fig 1. Nicotine increased the coupling of DMP and reversals. (A)** An assay diagram of response to nicotine exposure and withdrawal in *Caenorhabditis elegans*. Young adult hermaphrodites were used. **(B)** Representative 3-min locomotion trajectories of two animals on withdrawal plates without

(− nic) and with (+ nic) nicotine exposure. **(C)** Coupling time diagrams for reversal and defecation motor program (DMP). Upper, representative speed traces labeled with the moment of DMP setup. Gray bar: DMP not couple with reversal; Orange bar: DMP couples with reversal. Bottom, raster plots of 10 consecutive DMP cycles that coupled with or without reversal in 10 animals. **(D)** Quantification of the DMP-reversal coupling proportion. Nicotine, but not glutamate, significantly enables the coupling of DMP to reversal. Two-way ANOVA was performed (interaction: $F_{(1, 37)}$ = 225.9, $P < 0.0001$). **(E)** 100 segmented DMP-associated movements, each shown for 30 s. The 0-s time point represents the initiation of the DMP and serves as the alignment reference. The speed of 10 s before and 20 s after DMP initiation were used. The figure below shows the average speed of the 100 segments. **(F)** The dose dependence of nicotine (3 h) for the coupling of DMP and reversal. **(G)** The time dependence of nicotine (1 mM) for the coupling of DMP and reversal. **(H)** The failure time dependence of nicotine. $n \geq 10$ animals. ns, no significance, *** $p < 0.001$ by Two-way ANOVA analysis. Error bars, SEM. The data underlying this figure can be found in S1 Data.

These results indicate that intact pBoc activity is critical for nicotine-enhanced motor coupling, consistent with nicotine amplifying an interoceptive response to body wall contraction rather than altering DMP timing. In agreement, the delay from DMP onset to reversal initiation was unchanged by nicotine (S2F Fig).

To more precisely quantify the timing, we aligned all DMP events to the moment of minimal body length (time 0) and extracted locomotion speed traces spanning 10 s before to 20 s after this point (Fig 1Ei). This alignment allowed us to precisely synchronize locomotion speed at the moment of DMP, facilitating a comprehensive analysis of the locomotion characteristics associated with 100 DMP events across 10 worms. By aligning the data in this way, we ensured that variations in DMP cycle timing did not obscure the underlying coupling patterns, thereby improving the accuracy and reliability of our analysis. We found that nicotine induced a pronounced backward movement immediately following DMP compared to controls (Fig 1Eii).

Nicotine-enhanced DMP-reversal coupling was both concentration and exposure-time-dependent. With a fixed 3-h exposure, coupling increased with nicotine concentration, yielding a half-maximal effective concentration ($EC_{50}$) of 0.34 mM (Fig 1F). At a fixed saturated dose (1 mM), coupling increased with exposure duration, with a half-maximal effect at ~15.2 min (Fig 1G). Furthermore, similar to how nicotine induces a reversible aversive reaction in humans [34], nicotine-enhanced coupling was reversible: after transfer to nicotine-free plates, coupling gradually returned to baseline (~20%) with a half-life of ~1.2 h (Fig 1H).

Given that DMP is an intrinsic bodily rhythm, these findings suggest that nicotine acts as an interoceptive modulator, selectively enhancing sensory-motor linkage between internal rhythmic contractions and reversal behavior [35–37]. This prompted us to investigate the molecular and circuit-level mechanisms underlying this phenomenon.

## Nicotine-induced abnormal motor coupling requires the nicotinic acetylcholine receptor ACR-16

Previous studies have shown that nicotine-dependent behaviors require nAChRs [5,38], which mediate responses to the neurotransmitter acetylcholine (ACh) [39,40]. To test whether nAChRs mediate nicotine's effects in our assay, we pharmacologically blocked these receptors using DHβE, a competitive antagonist of nAChRs [40]. DHβE treatment significantly suppressed nicotine-enhanced DMP-reversal coupling (Fig 2A). Consistently, other nicotine-induced motor behaviors—including increased reversal frequency (S3A Fig), prolonged reversal duration (S3B Fig), and enhanced reversal and forward speeds (S3C and S3D Fig)—were also inhibited. In contrast, a cocktail of glutamate receptor antagonists, DNQX and MK-801, which block ionotropic excitatory glutamate receptors [41,42], did not affect the coupling proportion (Fig 2B). These pharmacological results provide strong evidence that nicotine's action in *C. elegans* depends on nAChRs rather than glutamate receptors.

To further validate this, we performed a genetic screen focusing on nAChR mutants. The *C. elegans* genome encodes over 29 nAChR subunits [43], many with available loss-of-function alleles (Fig 2C) [16,44,45]. Among these, only *acr-16* mutants completely lost the nicotine-enhanced DMP-reversal coupling, with nicotine failing to induce motor coupling (Fig 2D and S3 and S4 Movies). Other uncoordinated mutants such as *unc-29*, *unc-38*, and *unc-63* showed reduced coupling

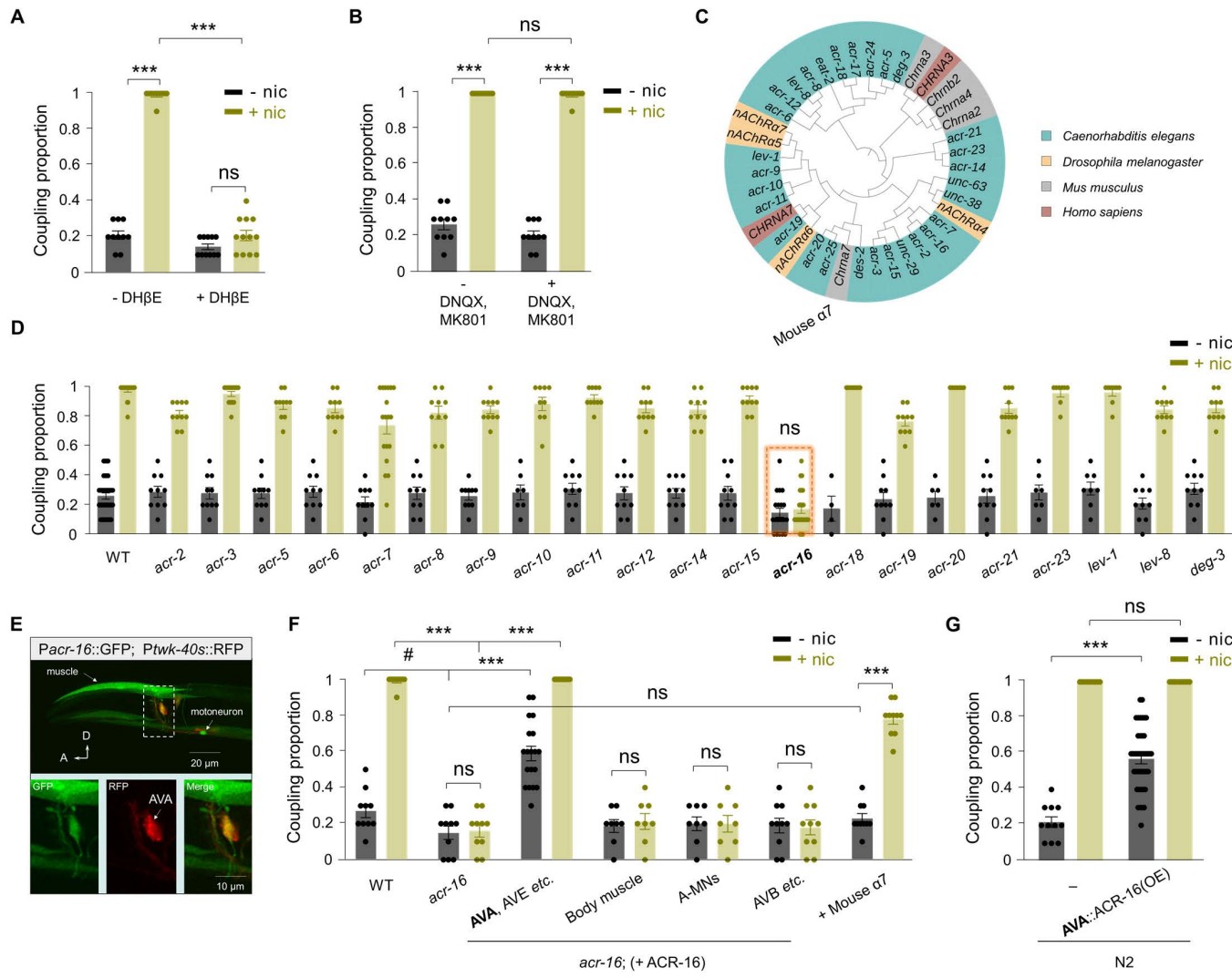

**Fig 2. Nicotine enhanced DMP-reversal coupling requires ACR-16 in AVA. (A)** Suppression of nicotine enhanced DMP-reversal coupling by nAChR antagonist DHβE (20 μM). Two-way ANOVA was performed (interaction: $F_{(1, 39)}$ = 275.6, $P<0.0001$). **(B)** The GluR antagonists DNQX (500 μM) and MK-801 (50 μM) do not affect nicotine-enhanced DMP reverse coupling. Two-way ANOVA was performed (interaction: $F_{(1, 36)}$ = 1.619, $P=0.2114$). **(C)** Dendrogram of *Caenorhabditis elegans* nAChRs. A fraction of nAChRs from *Drosophila*, mouse, and humans were included, such as mouse α7. **(D)** Genetic screening identified a specific nAChR gene, *acr-16*, disrupts nicotine-enhanced DMP-reversal coupling. Two-way ANOVA was performed (interaction: $F_{(21, 437)}$ = 12.95, $P<0.0001$). **(E)** Expression pattern of *acr-16*. *Up*, endogenous *acr-16* promoter driven GFP was observed in AVA command interneuron and body wall muscles. Scale bar, 20 μm. *Bottom*, the co-localization of P*acr-16*::GFP and P*twk-40s*::RFP in AVA neuron. Scale bar, 10 μm. Worm orientation, A anterior, D dorsal. **(F)** Quantification of the rescue of DMP-reversal coupling by expression of ACR-16 in different tissues. AVA, AVE, *etc.* command interneurons, but not motor neurons or muscles, are the critical functional cells for ACR-16. Expression of mouse α7 nAChR restored the DMP-reversal coupling proportion in *acr-16(ok789)* mutants. Two-way ANOVA was performed (interaction: $F_{(6, 129)}$ = 34.81, $P<0.0001$). **(G)** Overexpression of ACR-16 in wild-type N2 increases the DMP-reversal coupling proportion even without nicotine exposure. Two-way ANOVA was performed (interaction: $F_{(1, 70)}$ = 22.05, $P<0.0001$). $n \geq 10$ animals. ns, no significance, *** $p<0.001$ by Two-way ANOVA analysis. # $p<0.05$ by Student $t$ test. Error bars, SEM. The data underlying this figure can be found in S1 Data.

proportions (S3E Fig), although their severe locomotion defects may nonspecifically affect the behavior [46,47]. In agreement with the pharmacological data, mutants lacking ionotropic glutamate receptors like *glr-1*, *glr-2*, *glr-4*, *glr-5*, *nmr-1*, and *nmr-2*, displayed normal DMP-reversal coupling (S3F Fig).

Together, these findings demonstrate that nicotine-induced abnormal motor coupling specifically requires the nAChR subunit ACR-16 in *C. elegans*, highlighting its pivotal role in mediating nicotine's modulatory effects on sensory-motor integration.

### ACR-16 acts in command interneurons to mediate nicotine-induced motor responses

*acr-16* encodes an α7-like nAChR broadly expressed in both the nervous system and body wall muscles [44,45,48]. To assess how *acr-16* contributes to nicotine-induced motor behavioral changes in *C. elegans*, we examined its expression pattern and functional site of action. Using a fluorescence reporter driven by the endogenous *acr-16* promoter, we detected expression in both neuronal and muscular tissues (Fig 2E). Within the nervous system, *acr-16* is predominantly expressed in AVA and AVB command interneurons and in a subset of motoneurons, all of which are critical for regulating motor behaviors [45,49,50]. In mammals, nicotine's behavioral effects are primarily mediated by neuronal, rather than muscle, nAChRs [40]. To pinpoint the neuronal site of *acr-16* action, we performed cell-specific rescue experiments in *acr-16* mutants. Restoring ACR-16 expression in the backward command interneurons AVA and AVE using the *nmr-1* promoter fully rescued the nicotine-enhanced DMP-reversal coupling (Fig 2F). In contrast, expression in body wall muscles (*myo-3* promoter), backward motor neurons (*unc-4* promoter), or forward command interneurons such as AVB and PVC (*sra-11* promoter) failed to restore nicotine responsiveness. These results indicate that ACR-16 acts primarily in the backward command interneurons to mediate nicotine-induced motor coupling.

Interestingly, overexpression of ACR-16 in AVA not only rescued the nicotine response in *acr-16* mutants but also elevated baseline DMP-reversal coupling in the absence of nicotine (Fig 2F). This suggests that the expression level of ACR-16 in AVA may set the threshold for coupling. To test this, we overexpressed ACR-16 in AVA using the short *twk-40s* promoter [51] in wild-type N2 worms. Indeed, even without nicotine, these animals exhibited increased baseline coupling, and nicotine exposure drove the coupling proportion to 100% (Fig 2G).

Together, these findings demonstrate that ACR-16 mediates nicotine's behavioral effects primarily via backward command interneurons, and that the strength of this response is strongly dependent on ACR-16 expression levels in these neurons.

### The Mouse nAChR α7 subunit can functionally substitute for worm ACR-16

Given the essential role of ACR-16 in nicotine-dependent behavioral response in *C. elegans*, we asked whether the mammalian homolog could serve a similar function. ACR-16 shares high sequence similarity with the vertebrate α7 subunit of nAChRs (Fig 2C) [44,45]. To test cross-species functional compatibility, we expressed the mouse α7 nAChR in *acr-16* mutant worms under the control of the endogenous *acr-16* promoter. Remarkably, mouse α7 expression restored nicotine-enhanced DMP-reversal coupling in the mutants (Fig 2F). However, unlike worm ACR-16, the mouse α7 receptor did not increase baseline coupling proportion in the absence of nicotine (Fig 2F). These findings suggest that α7/*acr-16*-dependent nicotine responses are conserved between mammals and *C. elegans*.

### Nicotine enhances AVA calcium activity linked to the DMP

To establish the physiological role of AVA interneuron in nicotine's interoceptive effects, we measured real-time AVA activity in freely moving animals using the genetically encoded calcium ($Ca^{2+}$) indicator GCaMP6s, with wCherry as a ratiometric reference [52] (Materials and methods). These AVA::GCaMP transgenic animals responded to nicotine similarly to wild-type N2 worms in terms of reversal frequency, duration, and locomotion speed (wild-type, S4 Fig). In parallel, we used an independent infrared imaging system to monitor the rhythmic defecation behavior in real time [53] (Materials and methods). AVA $Ca^{2+}$ activity was recorded across five consecutive DMP cycles to assess their synchrony.

Following nicotine exposure, AVA displayed significantly elevated $Ca^{2+}$ oscillation compared to nicotine-free conditions (Fig 3A). Both the frequency and amplitude of $Ca^{2+}$ transients were markedly increased in the wild-type transgenic worms (Fig 3A—3D), indicating that nicotine enhances AVA excitability. This nicotine-induced AVA activation was strongly temporally coupled to DMP events. To precisely define this relationship, we aligned AVA $Ca^{2+}$ traces to the shortest body length point of each DMP cycle, extracting activity from 10 s before to 20 s after this reference point (Fig 3A). This alignment allowed us to synchronize AVA activity with 100 DMP events across 20 worms. Under nicotine treatment, nearly every DMP event was accompanied by a robust AVA $Ca^{2+}$ transient (Fig 3B and 3C), whereas in the absence of nicotine, DMP initiation elicited minimal AVA activity (Fig 3B). These results parallel the behavioral observation that nicotine enhances DMP-triggered reversals, suggesting that elevated AVA activity drives this coupling.

Consistent with behavioral requirements, AVA $Ca^{2+}$ activity was nearly absent in *acr-16* mutants under nicotine exposure (Fig 3B–3D). In these mutants, nicotine failed to increase either the response probability or the peak amplitude of AVA transients. Cell-specific rescue of ACR-16 in AVA using the short *twk-40s* promoter fully restored both the frequency and amplitude of DMP-evoked $Ca^{2+}$ transients (Fig 3B—3D). Notably, ACR-16 rescue also modestly increased baseline AVA activity in the absence of nicotine, mirroring the enhanced baseline DMP-reversal coupling seen in ACR-16-overexpressing worms. Furthermore, the ACR-16 dependence of nicotine-induced increases in reversal frequency, duration, and locomotion speed was confirmed in these rescue animals (S4 Fig). Together, these findings indicate that nicotine enhances motor coupling by elevating AVA activity in an ACR-16-dependent manner.

## Nicotine upregulates functional expression of ACR-16

Having established that enhanced AVA activity is crucial for nicotine's action and that this regulation depends on ACR-16, we next aimed to identify how ACR-16 mediates this activity. In mammals, chronic nicotine exposure is well known to up-regulate nAChRs, contributing to addiction and withdrawal responses [54,55]. Given the specific requirement of ACR-16 for both the behavioral and $Ca^{2+}$ responses we observed, we hypothesized that nicotine might increase the functional expression of ACR-16 in AVA, thereby driving the enhanced coupling.

To test this, we generated a transgenic *C. elegans* strain expressing functional ACR-16 cDNA tagged with a fluorescent marker (SL2d::GFP) in AVA neuron in wild-type N2 worms. Quantification of GFP fluorescence intensity before and after nicotine treatment revealed a significant increase following nicotine exposure (S5A and 5B Fig). To avoid potential artifacts arising from overexpression or non-specific promoters, we employed CRISPR-Cas9 genome editing to generate an AVA-specific ACR-16::GFP knock-in strain (EN9002) that preserves endogenous expression levels [50,56,57]. In this strain, an spGFP11 tag was inserted into the *acr-16* locus, spGFP1–10 was expressed specifically in AVA using a Cre-loxP-based promoter combination (Materials and methods). ACR-16::GFP signal was enriched in the AVA soma, with punctate labeling along the neurite (Fig 4A). Consistently, nicotine exposure significantly enhanced ACR-16::GFP fluorescence in both soma and neurite compartments (Fig 4A—4C), indicating that nicotine upregulates ACR-16 expression in AVA.

The behavioral, $Ca^{2+}$ imaging, and expression results all point to a potential mechanism in which nicotine upregulates ACR-16, thereby enhancing AVA activity and promoting reversals. To directly test this idea, we performed in situ whole-cell patch-clamp recordings from dissected AVA neurons to measure ACh-evoked currents (Fig 4D). In untreated wild-type animals, ACh (1 mM) elicited robust inward currents (Fig 4D), confirming the presence of functional AChRs in AVA. Strikingly, nicotine-pretreated animals displayed significantly larger ACh-evoked currents (Fig 4D and 4E). Application of the nAChR-selective antagonist DHβE strongly attenuated this nicotine-induced potentiation (S5C and S5D Fig). These results indicate that AVA not only expresses functional ionotropic AChRs [49], but that their expression may also be enhanced following nicotine exposure. In light of our previous findings identifying a specific ACR-16 dependence, we next examined whether ACR-16 was required for this potentiation. In *acr-16* mutants, nicotine exposure failed to increase ACh-evoked currents (Fig 4D and 4E). Reintroducing ACR-16 in AVA restored the nicotine-induced enhancement (Fig 4D and 4E). These findings suggest that nicotine-enhanced ACh current in AVA is also mediated by ACR-16.

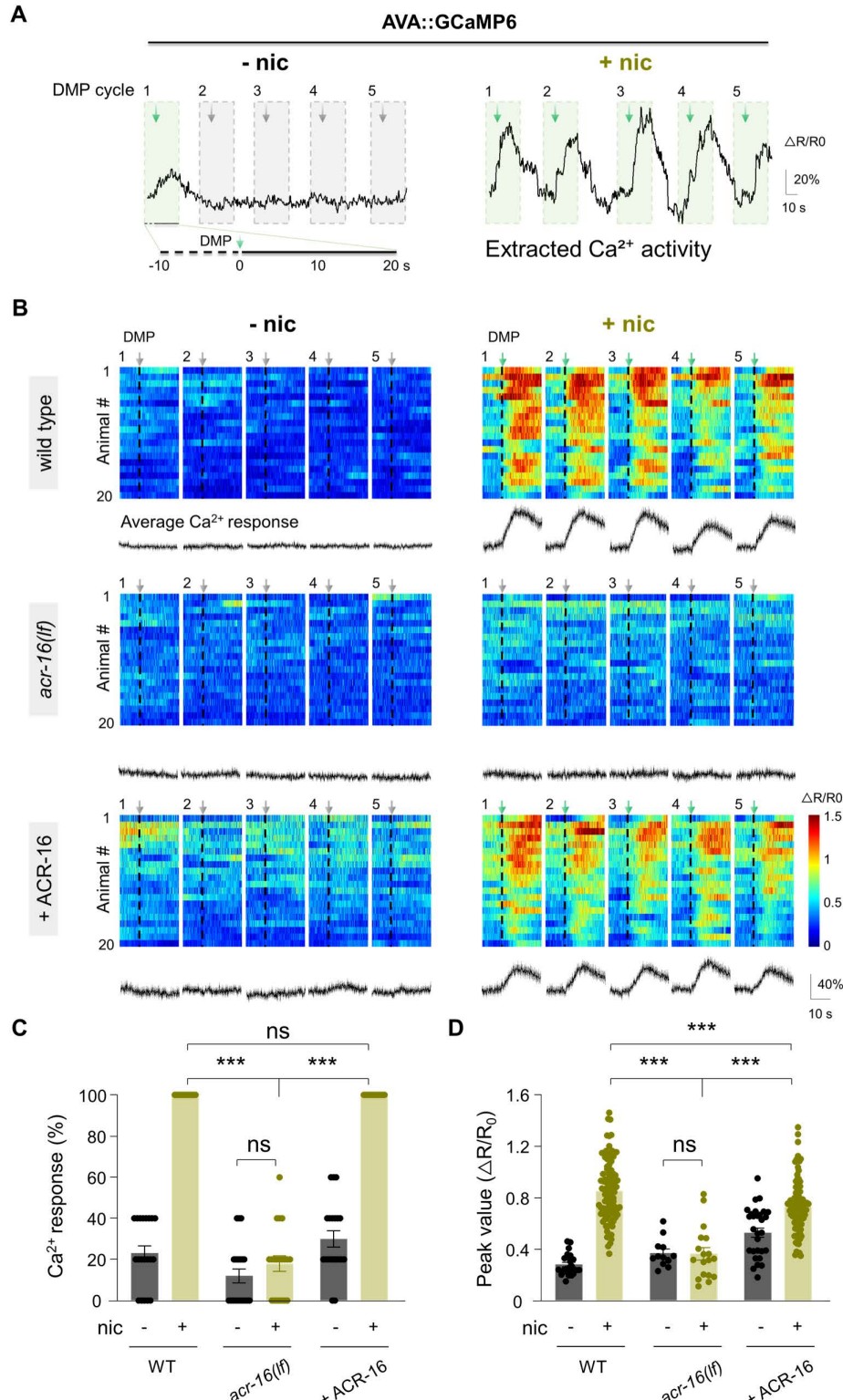

**Fig 3. Nicotine enhances calcium activity in AVA. (A)** *Left*, A representative calcium (Ca²⁺) transient of AVA neurons over five consecutive DMP cycles from a freely behaving animal without nicotine treatment. Only one DMP triggers Ca²⁺ activity in AVA. *Right*, Nicotine significantly enhances spontaneous Ca²⁺ transient activity in AVA, with all five DMP cycles initiating Ca²⁺ activity in AVA. Green boxes and arrows indicate tight coupling between DMP and

AVA Ca$^{2+}$ transients, while gray boxes and arrows represent the absence of coupling. Dashed boxes highlight 30 s segments of AVA Ca$^{2+}$ activity, with 10 s before and 20 s after DMP initiation. Arrows indicate the initiation moments of DMP. **(B)** The AVA Ca$^{2+}$ activity from five consecutive DMP cycles, recorded from 20 individual worms, is shown for both nicotine-free (− nic) and nicotine-exposed (+ nic) conditions across different genotypes (n = 20 animals per genotype). The Ca$^{2+}$ transients are aligned based on the DMP initiation moments (indicated by arrows and dashed lines). Numbers 1 through 5 correspond to the five consecutive DMP cycles. The average AVA Ca$^{2+}$ response for each DMP is displayed below. **(C)** Quantitative analysis of the coupling percentage of AVA Ca$^{2+}$ transient responses to DMP. Nicotine enhanced coupling proportion of DMP-AVA Ca$^{2+}$ transients is dependent on ACR-16. Two-way ANOVA was performed (interaction: $F_{(2, 114)}$ = 83.67, $P < 0.0001$). **(D)** Quantification of the peak amplitude of AVA Ca$^{2+}$ transients that coupled with DMP. Nicotine increased the amplitude of AVA Ca$^{2+}$ transients, which is dependent on ACR-16. Two-way ANOVA was performed (interaction: $F_{(2, 277)}$ = 26.60, $P < 0.0001$). n = 20 animals in each genotype. ns, no significance, *** $p < 0.001$ by Two-way ANOVA analysis. Error bars, SEM. The data underlying this figure can be found in S1 Data.

Notably, in nicotine-free conditions, *acr-16* mutants exhibited smaller ACh currents than wild-type worms, and this deficit was rescued by AVA-specific ACR-16 expression—consistent with the reduced DMP-reversal coupling observed in these mutants in the absence of nicotine (Fig 4D and 4E). Importantly, neither nicotine exposure, *acr-16* mutation, nor DHβE application significantly altered AVA resting membrane potential or voltage-dependent conductance (S6 Fig), suggesting that the enhanced Ca$^{2+}$ activity arises from increased synaptic input rather than changes in intrinsic excitability.

### Touch receptor neurons mediate nicotine-induced abnormal motor coupling

What signaling pathway links the DMP-reversal coupling with enhanced AVA Ca$^{2+}$ activity? The *C. elegans* DMP involves rhythmic release of transmitters (e.g., H$^+$) and peptides (e.g., NLP-40) from the intestine, which trigger strong contractions of body wall and enteric muscles [32,58]. One possibility is that these signaling molecules directly stimulate AVA, leading to enhanced activity. However, the receptors for H$^+$ and NLP-40 are neither expressed nor functional in AVA [32,59], making direct activation unlikely. An alternative hypothesis is that the forceful muscle contractions during DMP generate mechanical stimuli that activate mechanosensory, proprioceptive, or nociceptive neurons, which in turn influence AVA activity. To test this, we systematically screened a range of sensory-deficient mutants. Worms lacking sensory receptors in nociceptive neurons (ASH, OLQ, ALA, IL1), including *deg-1*, *tmc-1*, *trpa-1*, and *osm-9* [60–62], displayed normal nicotine-enhanced DMP-reversal coupling (Fig 5A). Similarly, mutants deficient in major proprioceptive receptors (*trp-1*, *trp-2*, *trp-4*) expressed in DVA [16,63] or SMD neurons [63] also retained robust coupling in response to nicotine (Fig 5A). These results suggest that nociceptive and proprioceptive neurons are not required for the effect.

Strikingly, mutants lacking key components of the gentle-touch mechanosensory channel complex—*mec-2*, *mec-4*, or *mec-6*—exhibited a profound loss of nicotine sensitivity, such that nicotine exposure no longer induces significant DMP-reversal coupling (Fig 5A). These *mec* genes encode essential mechanosensory channel complexes in TRNs (ALM, AVM, PLM, and PVM) [64] and did not significantly affect the DMP cycle length (S7A Fig). Furthermore, analysis of *mec-4(e1611)* mutants showed no significant changes in reversal frequency under either on-food or off-food conditions (S7B Fig) [65], indicating no locomotory defects that would interfere with the analysis of motor coupling. In *mec-6* mutants, reintroducing MEC-6 specifically into TRNs fully restored nicotine-enhanced coupling (Fig 5A). By contrast, other mechanosensory neurons, such as PVD and FLP, which mediate harsh-touch sensation and are affected by *mec-10* mutations, showed no changes in coupling (Fig 5A). Together, these results indicate that mechanosensory input from TRNs is essential for nicotine's effect on abnormal motor coupling.

To further confirm this, we performed optogenetic inhibition of TRNs using GtACR2, a light-gated inhibitory chloride channel [66,67]. Light illumination completely suppressed nicotine-enhanced DMP-reversal coupling (Fig 5B). Likewise, targeted ablation of TRNs using neuron-specific expression of miniSOG (mini singlet oxygen generator) [68] (S7C Fig)—which induces acute functional loss and subsequent neuronal death upon photoactivation—abolished the nicotine effect (Fig 5C). In contrast, ablation of the proprioceptive neuron DVA, previously reported to sense the body contraction [69], had no effect on the coupling (S7D Fig). These results demonstrate that TRNs are indispensable for mediating nicotine's action.

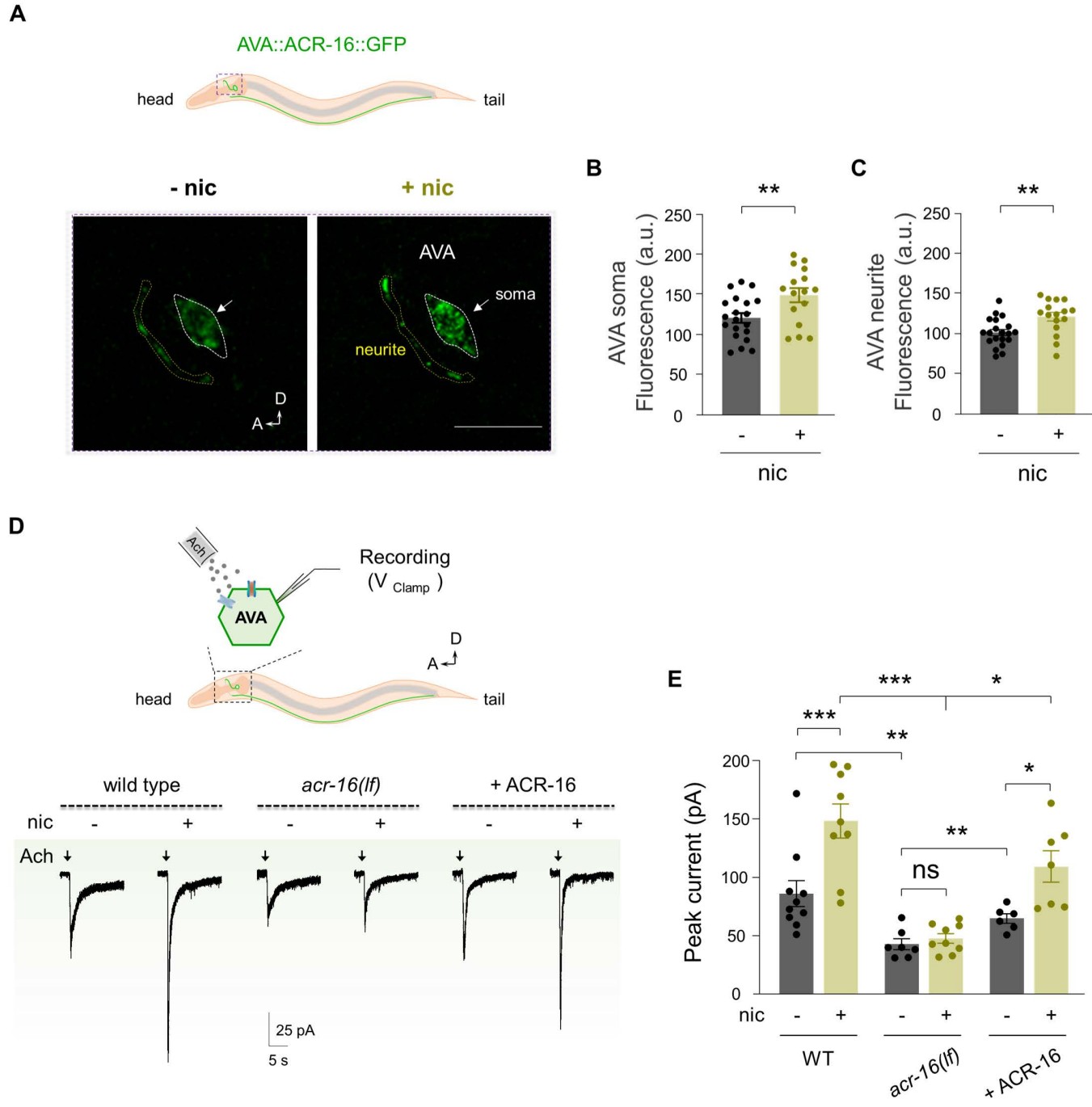

**Fig 4. Upregulation of ACR-16 is responsible for the nicotine-enhanced DMP-reversal coupling. (A)** Nicotine up-regulates ACR-16 expression in AVA. Using fluorescently-tagged ACR-16 based on split GFP reconstitution, we inserted three copies of the spGFP11 (split GFP11) sequence into the *acr-16*::aid::Scarlet locus via CRISPR/Cas9 (ACR-16-AID-Scarlet-spGFP11) and expressed the complementary spGFP1-10 (split GFP1-10) moiety in the AVA to generate AVA specific ACR-16::GFP transgenic line. Scale bar, 10 μm. **(B, C)** After nicotine exposure, the fluorescence intensity in both AVA soma and neurite are significantly increased. **(D)** *Up*, A schematic diagram of in situ whole-cell recording of *Caenorhabditis elegans* AVA under voltage clamping configuration ($V_{Clamp}$). *Bottom*, Representative currents evoked by acetylcholine (1 mM) ($I_{ACh}$) in different genotypes without (− nic) or with (+ nic) nicotine exposure. The purple arrows indicate the dosing time. AVA was hold at −60 mV. **(E)** Quantification of peak currents across different genotypes. Two-way ANOVA was performed (interaction: $F_{(2, 42)}$ = 4.105, $P = 0.0235$). $n \geq 6$ animals. ns, no significance, * $p < 0.05$, ** $p < 0.01$, *** $p < 0.001$ by Two-way ANOVA analysis. Error bars, SEM. The data underlying this figure can be found in S1 Data.

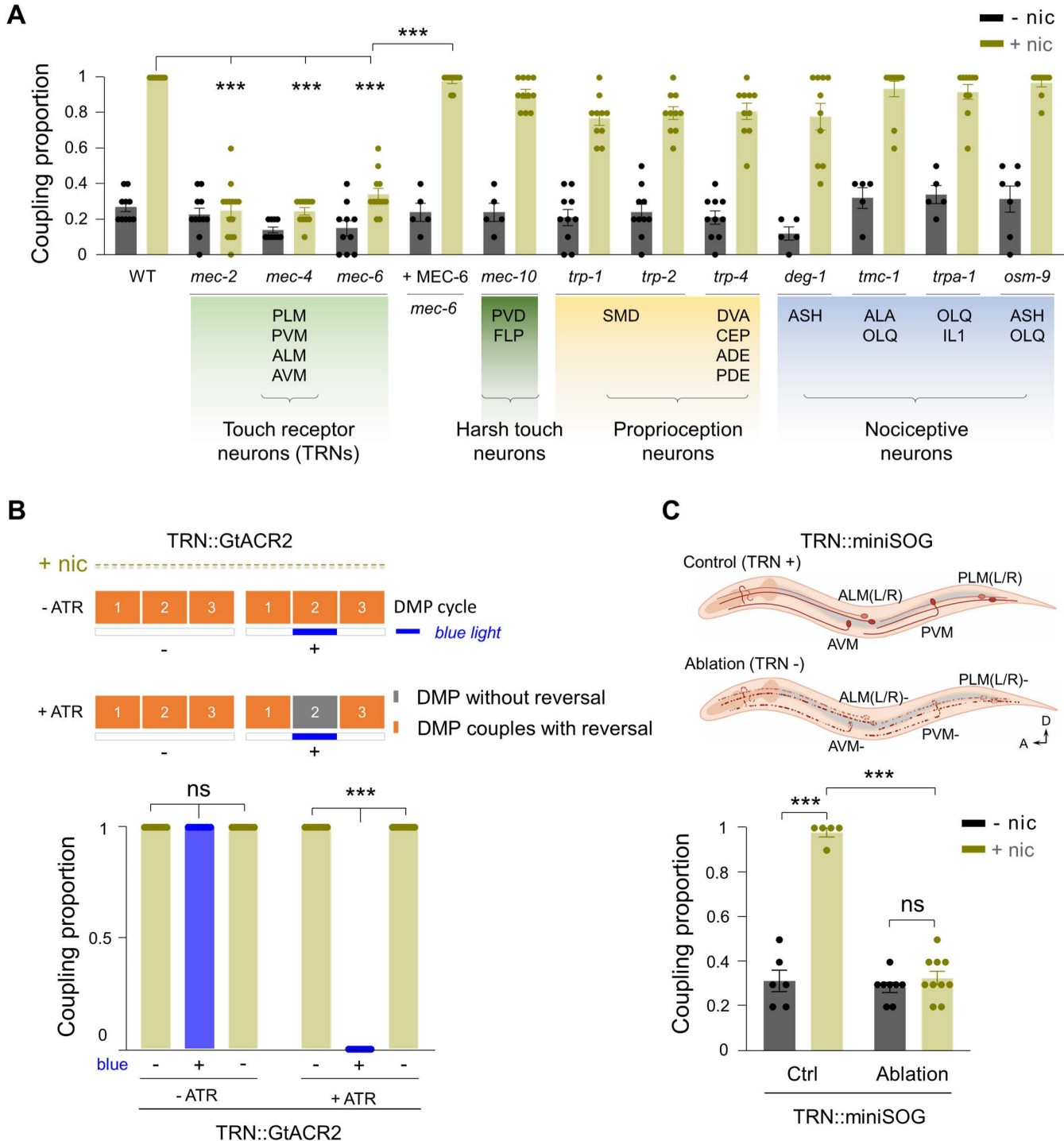

**Fig 5. Exteroceptive TRNs are essential for nicotine enhanced DMP-reversal coupling. (A)** Suppression of nicotine enhanced DMP-reversal coupling in *mec-2*, *mec-4*, *mec-6* mutation worms. Neuronal specific expression of MEC-6 restored nicotine enhanced DMP-reversal coupling in *mec-6* mutants. The coupling proportion was unaffected by mutations in genes essential for harsh touch, proprioception, and nociceptive functions. Two-way ANOVA was performed (interaction: $F_{(12, 217)}$ = 19.72, $P < 0.0001$). **(B)** *Up*, Blue light irradiation of TRNs expressing GtACR2 completely blocked nicotine-enhanced DMP-reversal coupling. The numbers 1 through 3 refer to three consecutive DMP cycles. Gray bar: DMP not couple with reversal; Orange bar: DMP couples with reversal. *Bottom*, Quantification of the proportion in transgenic TRN::GtACR2 animals without and with blue light after nicotine exposure. **(C)** *Up*, A schematic diagram of ablation of TRNs by miniSOG. *Bottom*, TRNs injury abolished nicotine enhanced DMP-reversal coupling. Two-way ANOVA was performed (interaction: $F_{(1, 25)}$ = 87.52, $P < 0.0001$). $n \geq 5$ animals. ns, no significance, *** $p < 0.001$ by Two-way ANOVA analysis. Error bars, SEM. The data underlying this figure can be found in S1 Data.

## Nicotine enhances the reversal sensitivity to gentle touch

TRNs are well-characterized mechanosensory neurons responsible for detecting exteroceptive gentle touch stimuli [70]. The finding that TRNs mediate nicotine-enhanced interoceptive response raises the question of whether nicotine might also affect exteroceptive sensitivity. To examine this, we applied gentle-touch stimulation to the anterior body using an eyelash probe [71]. Wild-type animals exhibited rapid and sustained reversals to touch, with an average reversal speed of 0.104±0.007 mm/s and a reversal duration of 4.9±0.4 s (Fig 6A—6C). Remarkably, in nicotine-treated worms, the touch elicited a faster reversal speed (+ nic, 0.136±0.006 mm/s) (Fig 6A and 6B), and a prolonged reversal duration (+ nic, 6.4±0.5 s) (Fig 6A and 6C).

Consistently, when we examined the gentle touch-evoked reversal response in *acr-16* mutants, nicotine exposure failed to enhance reversal speed (− nic, 0.096±0.006 mm/s; + nic, 0.098±0.005 mm/s) or duration (− nic, 4.3±0.3 s; + nic, 4.6±0.3 s) (Fig 6A—6C). Restoring *acr-16* expression in AVA rescued the nicotine-induced enhancement of reversal responses (Fig 6A—6C). These results demonstrate that nicotine not only induces an anxiety/aversion-like motor coupling behavior but also heightens escape responses to mechanical stimulation via ACR-16 in AVA. Both behaviors are regulated by TRNs, highlighting the critical role of TRNs in integrating nicotine's effects on abnormal sensory-motor coupling.

## Nicotine enhances neural activity from TRNs to AVA via ACR-16

We next asked how TRNs regulate nicotine-modulated behaviors. Connectome analyses indicate that TRNs form functional neural connections with command interneurons to mediate escape responses [72]. To assess the impact of nicotine on the neural activity between TRNs and AVA, we combined optogenetics together with in vivo electrophysiological recordings [73–75]. Light activation of TRNs (TRN::ChR2) with expressing the excitatory Channelrhodopsin-2 (ChR2) [76], evoked excitatory postsynaptic potentials (EPSPs) in AVA (Fig 6D). Nicotine exposure significantly increased EPSP amplitude (Fig 6D and 6E). Notably, this increase was blocked by the nicotinic receptor antagonist DHβE (Fig 6D and 6E). These results demonstrate that nicotine enhances the signal strength from TRNs to AVA.

To determine whether ACR-16 is required for nicotine-induced signal potentiation between TRNs and AVA, we analyzed the light-evoked EPSPs in *acr-16* mutants. Nicotine failed to further enhance EPSPs in *acr-16* mutants (Fig 6F and 6G). Reintroducing *acr-16* into AVA restored nicotine-enhanced EPSP potentiation. These results denote that nicotine-potentiated EPSPs depend on ACR-16 (Fig 6F and 6G). Interestingly, in the absence of nicotine exposure, EPSPs were reduced in *acr-16* mutants compared to wild-type animals (Fig 6G), suggesting that ACR-16 contributes intrinsically, at least in part, to basic neural signal from TRNs to AVA. This is consistent with our observation that ACh perfusion-evoked currents also require ACR-16.

In summary, our behavioral and in vivo electrophysiological data demonstrate that nicotine strengthens TRNs and AVA synaptic signal via ACR-16. This synaptic enhancement provides a neuronal integration mechanism for nicotine's modulation of both interoceptive and exteroceptive responses.

## Discussion

Nicotine exposure and withdrawal are known to induce a spectrum of emotional and sensory disturbances, yet the underlying mechanisms remain incompletely understood. In this study, we identify an unexpected neural pathway mediating high-dose nicotine-induced anxious/aversive-like reversal in *C. elegans*. We show that exteroceptive TRNs sense internal DMP, which in turn activate command interneurons to regulate the nicotine-induced behavioral response. Both TRNs mechanosensory function and the nicotine-dependent upregulation of the ACR-16 nAChR in AVA neurons are essential for this process (Fig 7). Moreover, nicotine potentiates signal strength between TRNs and AVA, revealing a striking form of sensory-motor coupling that links interoceptive and exteroceptive pathways at the circuit level. These findings broaden the current view of interoceptive processing, which is often considered to operate through dedicated, isolated ascending pathways.

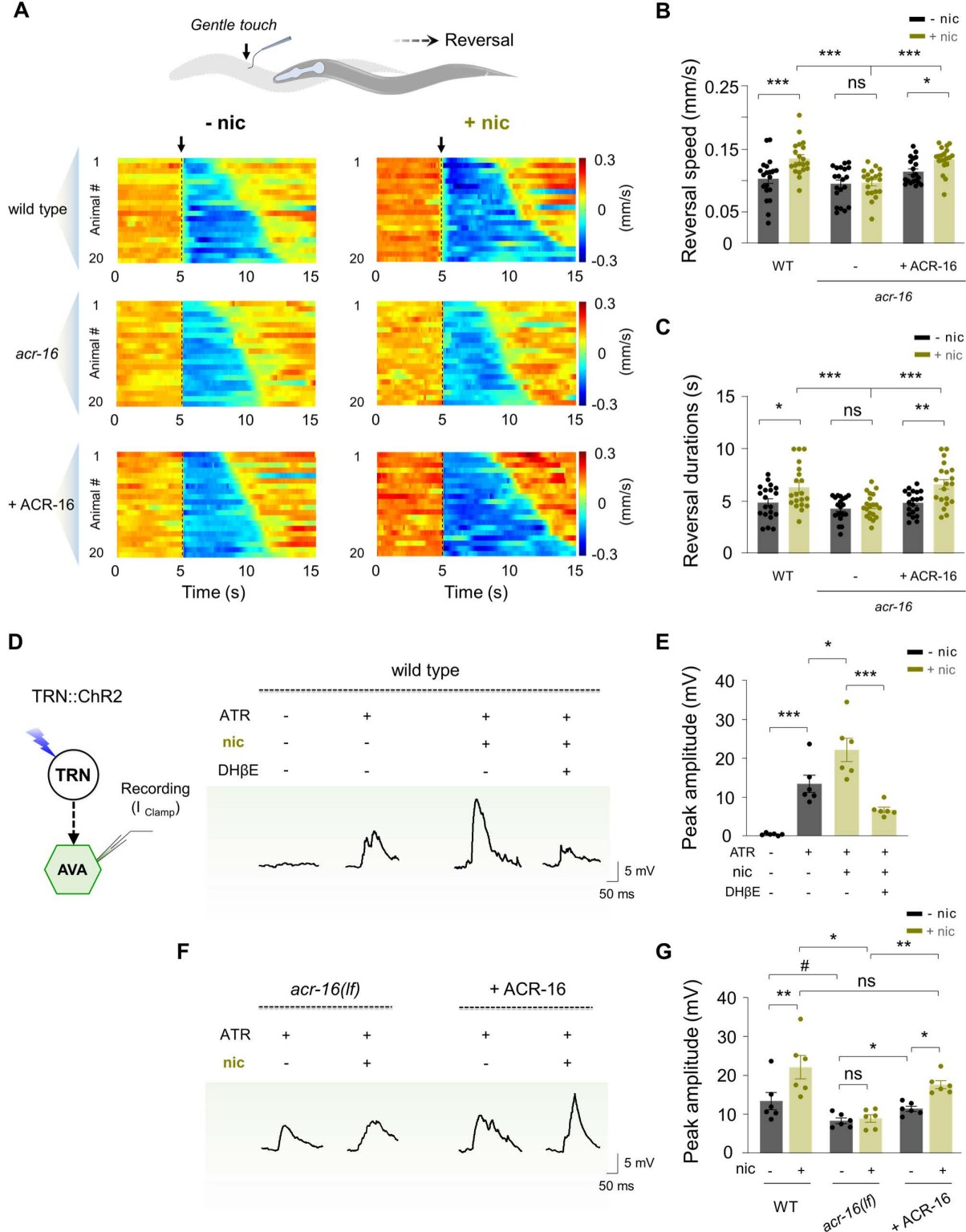

**Fig 6. Nicotine facilitates the signal strength from TRNs to AVA. (A)** *Upper*, a brief diagram shows standard gentle touch evoked reversal for extero-ceptive sensitivity examination. *Bottom*, raster plots showing individual speed responses of different genotypic animals to gentle touch on the anterior body, with and without nicotine exposure (− nic/ + nic) (*n* = 20 animals per group). Arrows denote gentle touch. **(B, C)** Quantification of the gentle touch

evoked reversal speed (B) and duration (C). Two-way ANOVA was performed (interaction: $F_{(2, 114)}$ = 3.557, $P$ = 0.0317 for (B); $F_{(2, 114)}$ = 2.581, $P$ = 0.0801 for (C)). **(D)** *Left,* schematic diagram of in situ whole cell recording of dissected AVA neuron by optogenetically stimulating TRNs under current clamp configuration ($I_{Clamp}$). *Right*, light-evoked excitatory post-synaptic potentials (EPSPs) under different pharmacological conditions in wild-type animals. **(E)** Quantification of the peak amplitude of the EPSPs. Nicotine significantly increased the signal strength in TRN-AVA circuit. ns, no significance, * $p < 0.05$, *** $p < 0.001$ by One-way ANOVA analysis ($F_{(3, 20)}$ = 23.87, $P < 0.0001$). Error bars, SEM. **(F)** TRNs evoked EPSPs in *acr-16* mutants and rescue lines. **(G)** Quantification of the peak amplitude of the EPSPs in different genotypes. Nicotine increased EPSPs were reduced in *acr-16* mutant animals and could be restored by expression ACR-16 back to AVA. Two-way ANOVA was performed (interaction: $F_{(2, 30)}$ = 3.067, $P$ = 0.0614). $n \geq 6$ animals. ns, no significance, * $p < 0.05$, ** $p < 0.01$, *** $p < 0.001$ by Two-way ANOVA analysis. # $p < 0.05$ by Student $t$ test. Error bars, SEM. The data underlying this figure can be found in S1 Data.

Nicotine is well established to modulate internal bodily sensations such as heart rate, blood pressure, and respiratory depth in mammals [77–79]. Here, we demonstrate that high-dose nicotine amplifies an apparent interoceptive signal in *C. elegans* by sensitizing the exteroceptive mechanosensory circuit. A similar nicotine-induced enhancement of mechanosensitivity has been reported in mice, where nicotine sensitizes cutaneous C-fiber sensory nerves [80]. While the contribution of such exteroceptive sensitization to mammalian interoception remains unclear, these parallels underscore a potentially conserved mechanism across species [81,82].

*C. elegans* has been reported to exhibit remarkable sensitivity to internal body states, such as muscle contractions [28,83], arousal levels [84], and intestinal activity [85,86]. Consistent with the earlier observations linking DMP to directed locomotion [28,83], our results suggest that somatic muscle contractions during the DMP can trigger reversals, and that nicotine markedly amplifies this coupling—from ~20% under baseline conditions to nearly 100% following exposure. The mechanism underlying nicotine-enhanced motor coupling appears to differ from spontaneous motor coupling. This is due to the following observations: (1) TRNs ablation does not reduce spontaneous DMP-reversal coupling; (2) spontaneous coupling persists in TRNs mechanosensory receptor-deficient *mec-2*, *mec-4*, and *mec-6* mutants; and (3) in *acr-16* mutants, spontaneous coupling remains present, albeit reduced, and is unaffected by nicotine. These distinctions indicate that nicotine does not simply amplify existing circuit output but instead recruits a distinct modulatory mechanism—one that could serve as a selective therapeutic target for nicotine-related behaviors without disrupting endogenous motor coupling.

At the molecular level, our data reveal that high-dose nicotine-induced coupling critically depends on ACR-16 nAChR upregulation in AVA neurons. This is distinct from low-dose nicotine effects, which involve both ACR-15 and ACR-16 [16]. While changes in other nAChR subtypes cannot be excluded [17,21,87,88], our electrophysiological and imaging results demonstrate that functional ACR-16 is indispensable. Notably, expression of mouse α7 nAChR fully restored the nicotine-induced coupling defect in *acr-16* mutants, despite its inability to substitute for low-dose nicotine effects [16]. Given that α7 nAChRs mediate high-dose nicotine aversion in rodents [13,89], these findings suggest a conserved role for this receptor subtype in high-dose nicotine responses across phyla.

Our circuit analyses also raise mechanistic questions. Optogenetic activation of TRNs evoked EPSPs in AVA, which were enhanced by nicotine and blocked by DHβE, but not by glutamate receptor antagonists. Neither glutamate exposure, glutamate receptor antagonists, nor mutations in ionotropic glutamate receptors affected nicotine's effects, implying that the observed excitatory EPSPs in the TRN-AVA circuit are not glutamatergic. Since TRNs are glutamatergic and lack expression of cholinergic markers such as *cha-1* (choline acetyltransferase) and *unc-17* (ACh transporter) [90], direct cholinergic transmission from TRNs to AVA seems also unlikely. A more plausible model is that the TRN-AVA functions indirectly, requiring additional ACh-releasing interneurons to relay TRN activation to AVA. Based on anatomical connectivity, we propose that AVD and AVE, both of which are cholinergic interneurons that receive strong TRNs inputs and project to AVA [91,92]. Cooperative activation of AVA, AVD, and AVE—well-known premotor interneurons for backward locomotion—could robustly drive nicotine-enhanced reversals. Additional or parallel mechanisms may also contribute, including neuropeptidergic modulation from TRNs [49,93].

PLOS Biology

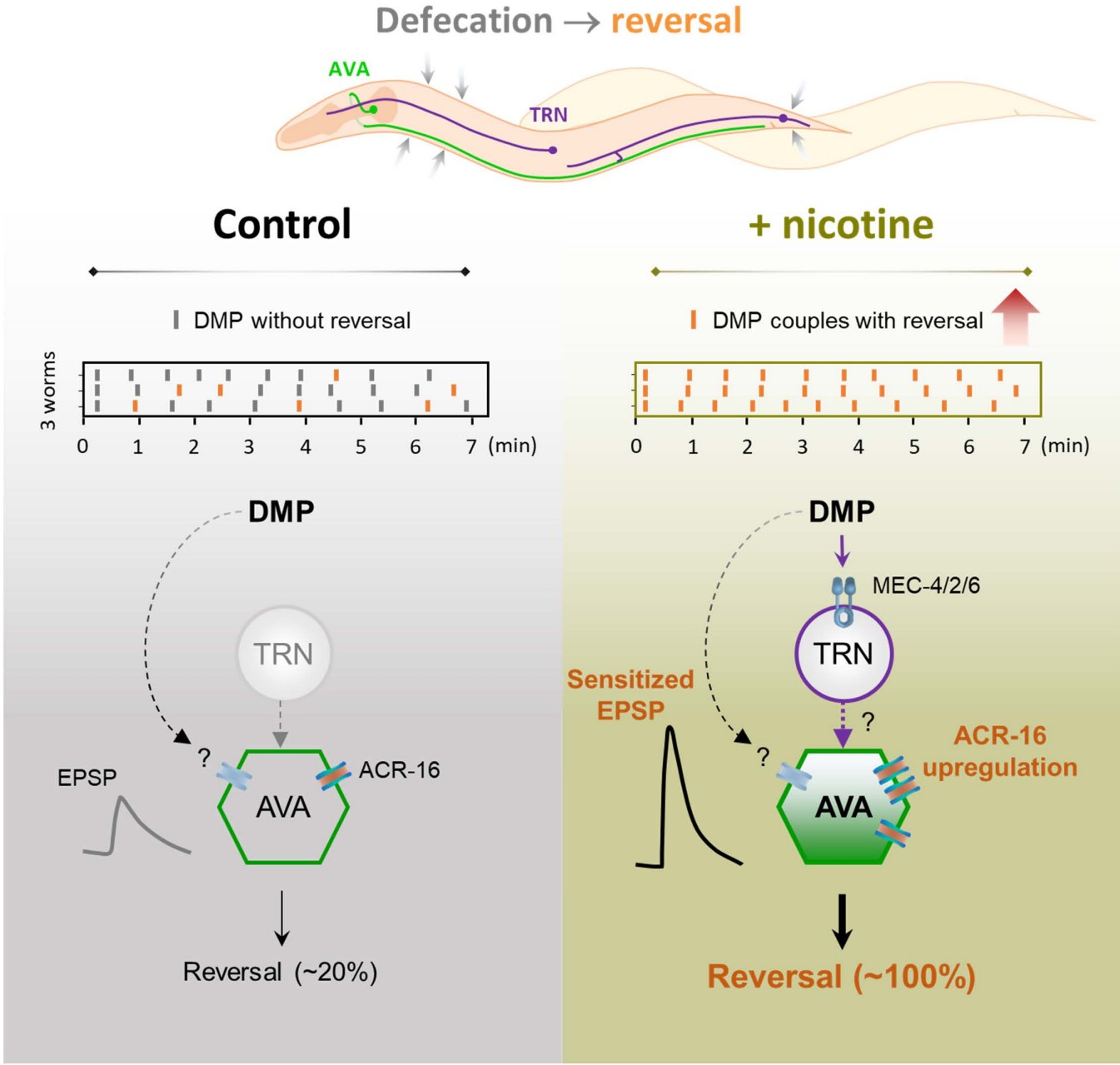

**Fig 7. Working model.** Nicotine significantly induces abnormal DMP-reversal coupling response in *Caenorhabditis elegans*. An unexpected mechanosensory neural pathway involving TRN-AVA circuit mediates this response. Key mechanisms include the nicotine-induced upregulation of ACR-16 nAChRs in AVA and nicotine-strengthened signal transmission within TRN-AVA circuit. TRNs may sense internal body contractions associated with the DMP, abnormally activating sensitized command interneurons to drive reversals. This finding underscores an unexpected functional integration induced by nicotine exposure.

Another open question is how TRNs detect DMP-associated muscle contractions. The spatial organization of TRNs suggests that posterior contractions should preferentially activate PLM and PVM, which typically promote forward movement, rather than the reversals we observed. This discrepancy implies the existence of alternative contractile signal

transmission routes or recruitment of other cholinergic neurons downstream of TRNs to bias AVA activation. Resolving these circuit-level details will be key to fully understanding nicotine's modulation of interoceptive–exteroceptive integration.

An important consideration in our study is that residual nicotine within the organism may constitute a lingering pharmacological state capable of directly influencing behavior. While we interpret these effects within a broad framework of interoception, nicotine may modulate motor coupling through multiple mechanisms—either via passive pharmacological action or by engaging specific internal sensors that shape interoceptive processing. Thus, nicotine-enhanced coupling likely reflects a combination of residual pharmacological effects and bona fide interoceptive sensing.

Finally, we find that nicotine upregulates ACR-16 expression in AVA under both heterologous (P*twk-40s*) and endogenous (P*acr-16*, CRISPR knock-in) promoter control. The similarity of these effects suggests a complex regulatory process that may involve both transcriptional and post-transcriptional mechanisms [94,95]. How does nicotine up-regulate ACR-16 expression at the cellular level? Nicotine may act as a molecular chaperone, enhancing the assembly, maturation, and trafficking of ACR-16 subunits, as reported in both heterologous systems and native neurons [96–98]. Another possibility is nicotine-mediated inhibition of receptor turnover, as observed for mammalian α4β2 receptors [99], suggesting that receptor accumulation may arise from reduced degradation. Determining the precise in vivo regulatory mechanism will be an important focus for future work.

## Materials and methods

### Maintenance and generation of *C. elegans* strains

All strains were maintained on standard NGM plates seeded with *E. coli* OP50, which were allowed to grow for at least 16 h prior to experimentation, and kept at 22 °C under standard conditions. Unless otherwise stated, "wild-type" refers to the Bristol N2 strain. For optogenetic experiments, strains were maintained in darkness on NGM plates supplemented with 0.5 mM all-trans retinal (ATR), with control groups reared on plates lacking ATR. Genetic mutants used to generate transgenic lines and compound mutants were obtained from the Caenorhabditis Genetics Center (CGC; http://www.cbs. umn.edu/CGC/) or the National Bio-Resources Project (NBRP; http://www.shigen.nig.ac.jp/c.elegans/indices.jsp).

Transgenic animals carrying non-integrated, extra-chromosomal arrays (*gaaEx*) were produced by co-injecting an injection marker along with one or more DNA constructs at 5–30 ng/μL. Stable integrated lines (*gaaIs*) were obtained from *gaaEx* strains via UV irradiation, followed by at least four rounds of outcrossing with N2. A complete list of strains used in this study is provided in S1 Table.

### Molecular biology

All expression plasmids were generated using the Multisite Gateway system (Invitrogen, Thermo Fisher Scientific, Waltham, MA, USA). Three entry clones—containing the promoter, target gene, and fluorescent marker (designated as slot1, slot2, and slot3, respectively)—were recombined into the pDEST R4-R3 Vector II via an LR reaction [100]. Genomic DNA from wild-type *C. elegans* was used to amplify *acr-16*, *twk-40s*, *nmr-1*, *mec-4*, *myo-3*, *sra-11*, and *unc-4*. The promoter lengths of P*acr-16*, P*twk-40s*, P*nmr-1*, P*mec-4*, P*myo-3*, P*sra-11*, and P*unc-4* were 3.5, 1.5, 5.0, 1.5, 2.3, 2.0, and 3.0 kb, respectively.

Genes of interest—including endogenous proteins (e.g., ACR-16) and exogenous functional proteins (e.g., GCaMP6s, GtACR2, ChR2, and miniSOG)—were inserted into *pDONR221* to generate slot2 constructs. Fluorescent proteins or the *unc-54*-3′UTR sequences were cloned into the *pDONR-P2R-P3* vector to generate slot3 constructs. The P*acr-16*::GFP reporter was generated using the Gateway recombination system (Invitrogen) in combination with In-Fusion cloning. To make the entry vectors, a 3.5 kb of sequence upstream of the *acr-16* gene was PCR amplified from *C. elegans* N2 genomic DNA using gene-specific promoter primers containing attB1 and attB2 sequences. The gene-specific sequences for P*acr-16* are as follows:

Forward: 5′-AGTTGCTCCCTCGAGACATTCACAACATATACGGTG-3′

Reverse: 5′-CTTGCACGACTAGTGGAGTGAATGTACATCCG-3′

The amplified promoter fragment was inserted into a linearized Gateway entry vector (Entry Clone A) using In-Fusion cloning, generating an entry clone containing the *acr-16* promoter flanked by attL1 and attL2 recombination sites. An LR recombination reaction was subsequently performed between the entry clone and a Gateway destination vector harboring GFP and the *unc-54*–3′UTR, resulting in the final expression construct, P*acr-16*::GFP::*unc-54*-3′UTR.

To visualize endogenous ACR-16 expression, three tandem copies of the spGFP11 sequence were inserted into the *acr-16::aid::Scarlet* locus using CRISPR/Cas9, generating *acr-16::aid::Scarlet::*spGFP11. Complementary spGFP1-10 was expressed specifically in AVA neurons. Scarlet fluorescence, visible with NeuroSIL, labeled all ACR-16 clusters, while GFP fluorescence was reconstituted exclusively in AVA neurons expressing spGFP1-10. AVA-specific expression was achieved using a Cre-loxP recombination strategy with P*flp-18* and P*gpa-14* promoters.

The strain driven by the P*twk-40s* promoter expressed ACR-16 cDNA tagged with GFP, resulting in predominant expression of functional ACR-16 in AVA neurons, as confirmed by GFP fluorescence. The cDNA sequence was amplified via 2× Phanta Max Master Mix (Dye Plus) Taq DNA Polymerase (Vazyme Biotech Co.). Detailed plasmid and primers information are provided in S2 and S3 Tables.

## Neuronal manipulation

To investigate the roles of TRNs and the proprioceptive neuron DVA in regulating DMP-reversal coupling, we expressed the genetically encoded photosensitizer miniSOG in these neurons driven by P*mec-4* and P*nlp-12* promoters, respectively. Neuronal ablation was achieved by exposing animals to 470 nm blue LED light (8.3 mW/cm$^2$) for 45 min. To verify the specificity and efficiency of ablation, cytoplasmic wCherry was co-expressed with miniSOG under the same promoter, enabling visualization of target neurons. Ablations were performed at the L2–L3 larval stage, and wCherry fluorescence was examined 24 h later to confirm loss of target neurons.

For optogenetic manipulation of TRN neurons, we used the neuron-specific promoter P*mec-4* to drive the expression of the GtACR2 or ChR2 in TRN neurons. Neuronal activity was inhibited or activated using blue light during behavioral assays. In GtACR2 experiments, TRN inhibition was used to evaluate the contribution of these neurons to nicotine-induced modulation of DMP behavior. Worms were subjected to three consecutive DMP trials: in the first and third trials (no blue light), TRNs were active, and nicotine-treated worms exhibited reversals following DMP. In the second trial, blue light was applied to inhibit TRNs, which abolished reversals after the DMP, indicating that TRN activity is required for nicotine-enhanced DMP-reversal coupling.

## Behavioral analysis

Behavioral assays were conducted at 22 °C. Late L4 stage or early young adults (12 h post L4 stage) hermaphrodites were used. 4.86 μL of nicotine (Shanghai PureOne Biotechnology, ʟ-Nicotine, HPLC purity ≥ 98%) was added into 1 mL of an *E. coli* OP50 suspension to create a 30 mM nicotine-OP50 mixture after vortexing for 1 min. 100 μL of this mixture was then dropped and spread onto the surface of a standard 3 mL NGM agar plate, and the NGM was shaken to ensure that the mixed solution evenly covered the entire surface. These nicotine plates were left at 22 °C for 16 h to allow diffusion of nicotine into the agar and OP50 bacterial lawn formation [101]. The final nicotine concentration in these plates is approximately 1 mM. NGM plates were prepared both with and without nicotine, with DHβE added to a final concentration of 20 μM. DNQX (500 μM) and MK-801 (50 μM) were simultaneously included in both nicotine-present and nicotine-absent NGM plates to serve as the treatment conditions for *C. elegans*. Worms were transferred to NGM plates with OP50 containing nicotine and cultured for 3 h prior to the experiment.

Defecation behavior was measured on NGM plates under standard conditions, as previously described [31]. The nematodes' defecation behavior was monitored for 10 consecutive cycles. The backward coupling rate was then calculated by dividing the number of backward events by the total number of defecations. The time window for assessing motor coupling was defined as the 5-s period following DMP initiation, during which reversals and calcium transients were recorded. Defecation phenotypes were scored under a modified stereomicroscope (Axio Zoom V16, Zeiss) equipped with a digital camera (acA2500-60 um, Basler).

For locomotion analysis, hermaphroditic worms, cultured under standard conditions, were transferred to imaging plates seeded with a thin layer of OP50 bacteria. One minute after the transfer, a 3-min video of the nematodes' locomotion was recorded. The eyelash touch assay was performed on food. Post-imaging analysis was conducted using an in-house developed MATLAB script, with the centerline used for tracking. The speed analysis of each animal was performed by dividing the recorded images into 33 segments, and the midpoint of each segment was used to calculate the velocity [100]. All images were captured using a 12× objective at a frame rate of 10 Hz.

## Confocal fluorescence microscopy

L4 stage transgenic animals expressing fluorescence markers were picked to imaging. Live worms were immobilized by 2.5 mM levamisole (Sigma-Aldrich) in M9 buffer. SGA741 gaaIs55 N2; [Ptwk-40s::ACR-16::GFP] fluorescence signals were captured using a Plan-Apochromatic 60× objective on a confocal microscope (FV3000, Olympus). EN9002 krSi81 [Pmyo-3::TIR1::bfp] I; bab535 [acr-16::aid::scarlet::spgfp11x3] V; krEx1404 [Pflp-18::lox::STOP::lox::spgfp1-10; Pgap-14::cre; Pmyo-2::mCherry] fluorescence signals were performed using an Andor Revolution XD laser confocal microscope system (Andor Technology plc., Springvale Business Park, Belfast, UK) based on a spinning-disk confocal scanning head CSU-X1 (Yokogawa Electric Corporation, Musashino-shi, Tokyo, Japan). The confocal system was mounted on an Olympus IX-71 inverted microscope (Olympus, Tokyo, Japan) and controlled by Andor IQ 1.91 software. The neurons and muscles labeled with GFP or RFP have been imaged with an excitation wavelength laser at 488 or 561 nm, respectively. The images were displayed using Image J 1.52o software (Wayne Rasband, National Institutes of Health, USA).

## Calcium imaging

The free-tracking Ca$^{2+}$ imaging setup consists of two integrated modules: a behavior tracking system and a fluorescence recording system. Worm behavior was imaged under dark-field illumination in the near-infrared range. Individual worms were picked from 6 cm NGM plates and placed on a THOPLABS SPW602 XY motorized stage for automated control. Imaging plates were prepared using the same agar composition as behavioral assay plates.

Behavioral tracking was performed using a 10× inverted objective (Olympus, Japan) and captured with a CCD camera (Point Gray Research, CM3-U3-13S2M, Canada). Custom real-time computer vision software kept the worm centered in the field of view by tracing the centerline of the worm. The fluorescence recording system records neuronal Ca$^{2+}$ activity using an sCMOS digital camera (Hamamatsu, Japan) equipped with a 10× objective lens (Olympus, Japan). To simultaneously capture images of wCherry and GCaMP6 side by side, we utilized a two-channel imager (W-VIEW GEMINI, Japan). Red and green channel images were simultaneously recorded at 10 frames per second (fps) with exposure times of 10 and 60 ms, respectively, using HCImage software (Hamamatsu).

Neural activity was quantified as the normalized change in the fluorescence ratio between GCaMP6s and wCherry, calculated as: $\Delta R/R_0 = (R - R_0)/R_0$, where $R = I_{GCaMP6s}/I_{wCherry}$, and $R_0$ was defined as the minimum $R$ value within the recording period. Intensities $I_{GCaMP6s}$ and $I_{wCherry}$ were extracted as the pixel intensity in the green and red channels, respectively, using custom MATLAB scripts. Analysis focused on AVA neuronal activity during freely moving worms performing the DMP.

A calcium response was defined as an increase in fluorescence intensity of ≥10% that occurred with a reversal behavior immediately following the pBoc phase of the DMP. Each such event was counted as one calcium response. We then calculated the calcium response percentage by dividing the number of these response events by five—the total number of

DMP cycles analyzed per animal. This quantification yields discrete values (e.g., 0, 20%, 40%, …, 100%) corresponding to the number of DMP cycles that showed a coupled calcium response. To minimize signal drift and motion artifacts, we limited our analysis to five consecutive DMP cycles per worm.

### Electrophysiology

Whole-cell patch-clamp recordings were performed following established protocols with minor modifications [102,103]. Briefly, 1–2-day-old hermaphrodite adults were immobilized on a Sylgard-coated (Sylgard 184, Dow Corning) coverslip using Histoacryl Blue adhesive (Braun) under a stereomicroscope (M50, Leica) and submerged in bath solution. Viscera were removed by gentle suction through a glass pipette, and the resulting cuticle flap was folded back and secured with WORMGLU adhesive (GluStitch) to expose the neuromuscular system.

Body wall muscle cells and neurons were patched using borosilicate glass pipettes (1B100F-4, World Precision Instruments) with tip resistances of 4–6 MΩ and 15–20 MΩ, respectively. Pipettes were pulled by micropipette puller P-1000 (Sutter) and fire-polished using an MF-830 microforge (Narishige). Recordings were obtained in the whole-cell configuration using an EPC9 amplifier (HEKA) controlled by Pulse software, with data processed using Igor Pro (WaveMetrics) and Clampfit 10 (Molecular Devices). Membrane currents were recorded at a holding potential of −60 mV, while membrane potentials were recorded at 0 pA. Signals were digitized at 10 kHz and low-pass filtered at 2.6 kHz.

The pipette solution contains (in mM): K-gluconate 115; KCl 25; $CaCl_2$ 0.1; $MgCl_2$ 5; BAPTA 1; HEPES 10; $Na_2ATP$ 5; $Na_2GTP$ 0.5; cAMP 0.5; cGMP 0.5, pH 7.2 with KOH, ~320 mOsm. cAMP and cGMP were included to maintain the activity and longevity of the preparation. The bath solution consists of (in mM): NaCl 150; KCl 5; $CaCl_2$ 5; $MgCl_2$ 1; glucose 10; sucrose 5; HEPES 15, pH 7.3 with NaOH, ~330 mOsm. All chemicals were purchased from Sigma unless otherwise specified. Experiments were conducted at 20–22 °C. To test AVA receptor function, ACh (1 mM in bath solution) was directly perfused onto patched neurons. For pharmacological blockade, recordings were performed in bath solution containing 20 μM DHβE. Optogenetic activation of TRNs during recordings was achieved by blue light illumination (470 nm, 13.75 mW/cm²).

### Statistical analysis and display

Statistical analyses were conducted via Student $t$ test, One-way ANOVA, or Two-way ANOVA analysis, two-sample $Z$ test, as appropriate. Data are presented as mean ± SEM, and significance thresholds were set at $p < 0.05$ (* $p < 0.05$, ** $p < 0.01$, *** $p < 0.001$). Calcium imaging data were visualized as heat maps (MATLAB, MathWorks), time-course curves (Igor Pro, WaveMetrics), and scatter plots (GraphPad Prism 9, GraphPad Software), with each point representing a single worm. Additional graphing software included Clampfit (Molecular Devices), ImageJ (NIH), and MEGA 6.60 (for phylogenetic analyses). For behavioral and fluorescence imaging experiments, unless otherwise noted, each recording trace represented an independent animal.

### Supporting information

**S1 Fig. Nicotine increases reversal frequency and speed. (A)** Raster plots displaying individual locomotion speed for animals without (− nic) and with (+ nic) nicotine exposure ($n = 22$ animals per group). **(B)** Quantification of the proportion change of forward and reversal movements with nicotine. Nicotine significantly increased the proportion of reversal. *** $p < 0.001$ by Two-sample Z test. **(C, D)** Quantification of the reversal frequency and duration changes after nicotine exposure. **(E, F)** Distribution of instantaneous speed of reversal (E) and forward (F) locomotion. Nicotine leads to a drastic increase of speed in both reversal and forward locomotion. *** $p < 0.001$ by Student $t$ test. Error bars, SEM. The data underlying this figure can be found in S1 Data.
(TIF)

**S2 Fig. Nicotine does not alter *Caenorhabditis elegans* DMP cycle. (A)** A schematic diagram of *C. elegans* defecation motor program (DMP). DMP is initiated by posterior body contraction (*pBoc*), and followed by anterior body contraction (*aBoc*) after ~2 s relaxation phase and then enteric muscle contraction, leading to expulsion of the gut contents (*Exp*), and an intercycle of approximately 45 s. **(B)** Representative ethograms of consecutive 10 defecation cycles in wild-type worms before (− nic) and after (+ nic) the exposure of nicotine (1 mM). Each dot represents 1 s. "p" stands for pBoc and "x" indicates Exp. aBoc is omitted due to difficulties in observation. **(C)** Quantification of the DMP cycle with or without nicotine exposure. **(D)** The distribution of the delay time between reversal initiation and different DMP phases. The aBoc is based on estimation. **(E)** Quantification of the average coupling events in wild-type and respective mutants in 10 min. Two-way ANOVA was performed (interaction: $F_{(4, 81)}$ = 15.82, $P < 0.0001$). **(F)** Quantification of the average delay time, which was not affected by nicotine exposure. $n \geq 8$ animals (Each dot represents a single DMP cycle). ns, no significance, * $p < 0.05$, ** $p < 0.01$, *** $p < 0.001$ by Two-way ANOVA analysis. Error bars, SEM. The data underlying this figure can be found in S1 Data.
(TIF)

**S3 Fig. Nicotine enhanced DMP-reversal coupling is independent on glutamate. (A)** Suppression of nicotine increased reversal frequency by DHβE (20 µM). Two-way ANOVA was performed (interaction: $F_{(1, 36)}$ = 34.69, $P < 0.0001$). **(B)** Suppression of nicotine increased reversal duration by DHβE (20 µM). Two-way ANOVA was performed (interaction: $F_{(1, 154)}$ = 5.442, $P = 0.0210$). **(C, D)** Distribution of instantaneous speed of reversal and forward locomotion. Nicotine leads to a drastic increase of speed in both reversal and forward locomotion (C). Velocity changes due to nicotine are all inhibited by DHβE (D). ns, no significance, ** $p < 0.01$, *** $p < 0.001$ by Student $t$ test. Error bars, SEM. **(E)** Mutations causing uncoordinated locomotion in nAChR mutants, including *unc-29*, *unc-38*, *unc-63*, suppressed the nicotine exposure induced DMP-reversal coupling. Two-way ANOVA was performed (interaction: $F_{(2, 29)}$ = 1.219, $P = 0.3101$). **(F)** No significant suppression of nicotine enhanced DMP-reversal coupling was observed in glutamate receptors mutant worms. Two-way ANOVA was performed (interaction: $F_{(5, 79)}$ = 2.669, $P = 0.0279$). $n \geq 5$ animals. ns, no significance, *** $p < 0.001$ by Two-way ANOVA. Error bars, SEM. The data underlying this figure can be found in S1 Data.
(TIF)

**S4 Fig. Nicotine-enhanced reversal in *hpIs580* was abolished in *acr-16* mutants. (A)** Quantification shows the nicotine enhanced reversal frequency observed in *hpIs580* (AVA::GCaMP) worms is abolished in *acr-16* mutants. Two-way ANOVA was performed (interaction: $F_{(2, 54)}$ = 17.92, $P < 0.0001$). **(B)** Quantification shows that the nicotine enhanced reversal duration observed in *hpIs580* worms is abolished in *acr-16* mutants. Reversal frequency and duration were restored by expression ACR-16 back. Two-way ANOVA was performed (interaction: $F_{(2, 274)}$ = 1.870, $P = 0.1561$). ns, no significance, ** $p < 0.01$, *** $p < 0.001$ by Two-way ANOVA analysis. Error bars, SEM. **(C, D)** Distribution of instantaneous speed of reversal (C) and forward (D) locomotion. Nicotine causes a sharp increase in the rate of reverse and forward motion in *hpIs580* worms, eliminated by *acr-16* mutants. $n = 10$ animals. ns, no significance, ** $p < 0.01$, *** $p < 0.001$ by Student $t$ test. Error bars, SEM. The data underlying this figure can be found in S1 Data.
(TIF)

**S5 Fig. Nicotine increases the expression of ACR-16. (A)** Nicotine up-regulates ACR-16 expression in AVA. Using ACR-16 cDNA with a fluorescent marker (sl2d::GFP) specifically in AVA neurons. **(B)** After nicotine exposure, the fluorescence intensity in AVA soma is significantly increased. **(C)** Nicotine-induced increase in I ACh could be suppressed by DHβE (20 µM). **(D)** Quantification of the peak current with DHβE. $n \geq 6$ animals. *** $p < 0.001$ by Student $t$ test. Error bars, SEM. The data underlying this figure can be found in S1 Data.
(TIF)

**S6 Fig. The membrane potential characteristics of AVA neurons are unaffected by nicotine and ACR-16. (A)** Resting membrane potentials (RMP) of all individual AVA neurons (gray lines) and the average RMP (black lines). **(B)** Quantification shows that the RMP of AVA exhibit no significantly change in different genotypes before and after nicotine exposure (1 mM, 3 h). Two-way ANOVA was performed (interaction: $F_{(2, 42)}$ = 0.3219, $P$ = 0.7265). ns, no significance, by Two-way ANOVA analysis. Error bars, SEM. **(C)** Representative step currents in AVA neurons from different genotypes. The currents were evoked by voltage clamp from −80 mV to + 80 mV at 20 mV increment. **(D)** Quantification of the I-V curve of the voltage-dependent currents. $n \geq 6$ animals. **(E–H)** DHβE (20 μM) had no effect on either the RMP (E, F) or the step currents (G, H) in AVA neurons. ns, no significance, by Student $t$ test. Error bars, SEM. The data underlying this figure can be found in S1 Data.
(TIF)

**S7 Fig. Ablation of DVA has no effect of nicotine's action. (A)** DMP cycle in wild-type N2 and *mec-2(e75)*, *mec-4(e1611)*, and *mec-6(e1342)* mutants. Two-way ANOVA was performed (interaction: $F_{(3, 712)}$ = 1.238, $P$ = 0.2948). **(B)** Quantification of the spontaneous reversal rate (# number of reversals per 3 min) in the presence (On food) and absence (Off food) of food, for wild-type (N2) and *mec-4(e1611)* mutants. Two-way ANOVA was performed (interaction: $F_{(1, 36)}$ = 0.08663, $P$ = 0.7702). **(C)** Representative TRNs (PLM) before and after the ablation. **(D)** Ablation of DVA did not change the nicotine enhanced DMP-reversal coupling proportion. Two-way ANOVA was performed (interaction: $F_{(1, 12)}$ = 0.1765, $P$ = 0.6818). ns, no significance, *** $p < 0.001$ by Two-way ANOVA analysis. Error bars, SEM. The data underlying this figure can be found in S1 Data.
(TIF)

**S1 Table. All *Caenorhabditis elegans* strains used in this study.** Genotypes used in each figure.
(PDF)

**S2 Table. The plasmids used in this study.** List of all plasmids used to generate transgenic lines.
(PDF)

**S3 Table. The primers used in this study.** Primers utilized for PCR amplification to identify mutants and construct transgenic strains.
(PDF)

**S1 Movie. Wild-type N2 movement without nicotine.** Wild-type N2 nematode movements during three consecutive DMP cycles without nicotine-exposure (N2, − nic).
(MP4)

**S2 Movie. Wild-type N2 movement with nicotine-exposure.** Wild-type N2 nematode movements during three consecutive DMP cycles with nicotine-exposure (N2, + nic).
(MP4)

**S3 Movie. *acr-16* movement without nicotine.** *acr-16* mutant movements during three consecutive DMP cycles without nicotine-exposure (*acr-16*, − nic).
(MP4)

**S4 Movie. *acr-16* movement with nicotine-exposure.** *acr-16* mutant movements during three consecutive DMP cycles with nicotine-exposure (*acr-16*, + nic).
(MP4)

**S1 Data. The data underlying the graphs shown in Figs 1–6 and S1–S7.**
(XLSX)

## Acknowledgments

We thank Tianqi Xu and Quan Wen for their assistance with data analysis. We thank Mei Zhen, Shawn Xu, Quan Wen, Lijun Kang, Zhuo Du, Xiao Liu, Haijun Tu for reagents and strains. We thank Mei Zhen, Quan Wen, Shawn Xu, and Lijun Kang for their valuable comments and discussion. We thank the *Caenorhabditis Genetics Center*, which is funded by the NIH Office of Research Infrastructure Programs (P40 OD010440, https://orip.nih.gov/), for strains.

## Author contributions

**Conceptualization:** Shangbang Gao.

**Funding acquisition:** Shangbang Gao.

**Investigation:** Yuting Liu, Leiru Huang, Ruipeng Wang, Fukang Qi, Yiwen Liu, Qingyuan Chen, Lili Chen, Shangbang Gao.

**Resources:** Morgane Mialon, Berangere Pinan-Lucarre.

**Supervision:** Shangbang Gao.

**Validation:** Shangbang Gao.

**Writing – original draft:** Shangbang Gao.

**Writing – review & editing:** Berangere Pinan-Lucarre, Shangbang Gao.

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
