## [Editor Report · Decision Letter 0]

1 May 2025

Dear Dr Gao,

Thank you for submitting your manuscript entitled "Nicotine induces abnormal motor coupling through sensitization of mechanosensory circuit in C. elegans" for consideration as a Research Article by PLOS Biology.

Your manuscript has now been evaluated by the PLOS Biology editorial staff, as well as by an academic editor with relevant expertise, and I am writing to let you know that we would like to send your submission out for external peer review.

Once your full submission is complete, your paper will undergo a series of checks in preparation for peer review. After your manuscript has passed the checks it will be sent out for review. To provide the metadata for your submission, please Login to Editorial Manager (https://www.editorialmanager.com/pbiology) within two working days, i.e. by May 03 2025 11:59PM.

Kind regards,

Taylor

Taylor Hart, PhD,

Associate Editor

PLOS Biology

thart@plos.org

---

## [Decision Letter · Decision Letter 1]

24 Jun 2025

Dear Dr Gao,

Thank you for your patience while your manuscript "Nicotine induces abnormal motor coupling through sensitization of mechanosensory circuit in C. elegans" was peer-reviewed at PLOS Biology. It has now been evaluated by the PLOS Biology editors, an Academic Editor with relevant expertise, and by several independent reviewers.

In light of the reviews, which you will find at the end of this email, we would like to invite you to revise the work to thoroughly address the reviewers' reports.

The reviewers have praise for the methodology, its technical merits, and the interest in the findings. However, they each noted various concerns, including some that are related to the unclear link circuit link between TRNs and AVA, missing controls related to the ACR-16::GFP reporter, and the possibility of nicotine dosing effects in addition to impacts on interoception. They also noted some other missing details and statistical issues. After discussing these points with the academic editor, we think that definitively determining the relay between TRNs and AVA may be beyond the scope of this study, but you should expand the discussion section to consider the possible intermediaries that may be transmitting the cholinergic signal. You should thoroughly address the other concerns raised by the reviewers, which will likely require new experiments.

Given the extent of revision needed, we cannot make a decision about publication until we have seen the revised manuscript and your response to the reviewers' comments. Your revised manuscript is likely to be sent for further evaluation by all or a subset of the reviewers.

**IMPORTANT - SUBMITTING YOUR REVISION**

*Re-submission Checklist*

*Published Peer Review*

*PLOS Data Policy*

*Blot and Gel Data Policy*

Sincerely,

Taylor

Taylor Hart, PhD,

Associate Editor

PLOS Biology

thart@plos.org

REVIEWS:

Reviewer #1: This study by Liu et al. investigates how nicotine exposure alters behavior in worms. The authors discover that nicotine significantly increases reversal frequency and duration during locomotion, and these reversals become tightly coupled with the defecation motor program (DMP). This behavioral change depends on heightened activity in the AVA command interneurons, mediated by upregulation of the nicotinic acetylcholine receptor ACR-16. Furthermore, the effect requires mechanosensory input from touch receptor neurons (TRNs). Functional assays demonstrate that nicotine-exposed animals exhibit enhanced synaptic strength in the TRN-to-AVA circuit, suggesting a novel form of sensitization. Mechanistically, the authors conclude that nicotine upregulates ACR-16 expression in AVA, increasing acetylcholine-evoked currents and AVA calcium activity. These changes sensitize AVA to mechanical input from TRNs. Genetic and optogenetic experiments confirm that both ACR-16 and functional TRNs are essential for nicotine-induced motor coupling. Overall, this is an intriguing study that reveals a surprising role for a classical touch circuit in mediating internal state changes following nicotine exposure. The methodology is rigorous. The conceptual link between nicotine, sensory-motor coupling, and interoception is thought-provoking.

In my opinion, however, there are a few gaps in the study that need to be addressed:

1) The central model proposed in this manuscript is that nicotine enhances AVA activity and reversal behavior through TRN-mediated mechanosensory input. However, the mechanism by which TRNs influence AVA remains unclear. As the authors themselves note, TRNs are glutamatergic neurons, while AVA responds to nicotine in an ACR-16-dependent, cholinergic manner. I appreciate that the authors suggest a cholinergic relay neuron may link TRNs to AVA, but no such neuron is identified or tested. Without direct evidence for this relay, the conclusion remains incomplete. This gap could significantly impact the interpretation of the main findings and should be addressed more directly.

Related to this, Figure 6D is confusing. If the effect is indirect, a solid arrow from TRNs to AVA may be misleading in the circuit diagram.

2) The finding that nicotine induces reliable reversal behavior tightly coupled to the DMP, especially during the pBoc phase, is intriguing and may reflect a form of interoceptive sensing of body contraction. However, the current data primarily establish a temporal correlation. Have the authors examined nicotine-induced reversal in egl-8, pbo-4, or pbo-5 mutants, which display disrupted pBoc? Additionally, what is observed in nlp-40 or aex-2 mutants, which disrupt DMP rhythmicity? These data could help distinguish whether nicotine-induced reversal depends on DMP timing, contraction, or both, and would strengthen the conclusion that mechanical signals associated with DMP are required for this behavior.

3) The authors report that nicotine exposure increases ACR-16::GFP fluorescence when the protein is expressed under the AVA-specific twk-40s promoter (lines 330-335). However, because this construct uses a heterologous promoter and acr-16 cDNA lacking endogenous introns or regulatory elements, it is difficult to interpret the fluorescence increase as transcriptional upregulation. Indeed, it is somewhat puzzling that this result appears consistent with the knock-in strain, where ACR-16 is expressed under its endogenous regulatory control. This raises questions about whether the observed effect reflects transcriptional regulation, protein stabilization, or some other mechanism. The authors should clarify this point.

Reviewer #2: The study by Liu and colleagues describes the effects of nicotine exposure on C. elegans behavior, at the neural circuit level. The work revealed that nicotine dramatically increases the coupling between the defecation cycle and reversing during locomotion, by enhancing the expression of a nicotine acetylcholine receptor in the AVA premotor neuron. Intriguingly, this coupling is mediated by the gentle touch mechanosensory neurons, know to respond to external mechanical stimulation, but now suggested to also detect internal movements associated with defecation, thus activating (directly or indirectly) the AVA neurons. The experiments appear well designed and performed, employing both standard methods and challenging techniques. The research is conveyed in a compelling and clear manner. The findings are of great interest to the understanding of how internal and external information may be integrated by the nervous system, and how a high dosage of nicotine may affect neural function. I have a few minor comments, but otherwise highly recommend the publication of this study.

Minor comments:

1. The term "intact" worms indicating non-exposed controls may be a little confusing. I suggest replacing it with "naïve", "control", or "non-exposed". In any case, after this term is introduced it is never used again, so it could also be removed altogether.

2. Fig. S1B a statistical test comparing proportions should be applied.

3. "These high-frequency, prolonged reversals, and high-speed triggered by nicotine resemble nicotine-induced anxious or aversive behaviors observed in mammals (25, 26), suggesting a conserved functional mechanism for nicotine response in C. elegans". I think this statement is too far reaching and circumstantial. I would leave out the second part of this claim, ", suggesting a conserved …".

4. Fig 1G instead of temporal dependence, it would be more precise to say dependence on exposure time. Temporal dependence may imply some kind of temporal pattern (e.g., on-off-on).

5. Not quite sure how the coupling proportion was calculated. According to the Methods it sounds like the number of reversals / calcium transients was divided by the number of defecation cycles. However, this does not necessarily imply synchrony. Was there a time window relative to each defecation cycle within which a reversal should have occurred? How was a calcium response vs. no response determined?

6. More details about the 2-way ANOVA results should be reported (i.e. the interaction term, the p-value etc.).

7. In Fig. 2E (bottom) it is worth labeling the separate and overlayed channels, just for clarity.

8. Fig. S4 is supposed to show normal nicotine behavioral responses in the GCaMP strain (line 286), but shows instead the effects of the acr-16 mutation (even though these were discussed earlier).

9. Regarding the acr-16:GFP strain, can you please provide more information in the Methods about how it was constructed, including relevant references?

10. I suggest to replace "neurotransmitter" with transmitter or signal when referring to H+ (line 377).

11. Fig. 3C unclear how the calcium response was quantified and why does it have discrete values.

12. In Fig. 5A the coupling is shown for the various mutants. Is the DMP cycle the same across mutants (especially TRN-related mutations)? For example, it has been shown that mec-4 have an altered the reversing frequency (both on and off food). How does this affect the coupling?

13. In Fig. 6D,E TRN-triggered AVA EPSPs are presented. In the Results this is inferred as TRN-AVA synaptic strength. However, as pointed out in the Discussion, this may likely be an indirect pathway, and so the phrasing should be changed in the Results and the Introduction.

14. The AVA rescue promoter (Ptwk-40) is not specific. When was the intersection promotor mentioned in the Methods used?

15. How was the eye lash touch assay performed? Was it on or off food?

16. The manuscript could benefit from overall further English editing.

Reviewer #3 [Vincent O'Connor]: The submission is an interesting study that provides a plausible mechanistic explanation for a nicotine induced behavioural response on motility and defecation. There is a detailed unpicking of the previously described loose coupling between these grosser behaviours. The text draws a lot of overviews from the mammalian literature subserved by the addictive issues of nicotine. I found this aspect a tad inflated and personally not required to support the value of the study. In addition, I think the grandiose against exteroceptive and interoceptive cueing of organismal behaviour difficult to follow. If the authors wish to pursue this tack, they should better justify it and address some of the clarifications that their study design raises. This is an interesting study, and one must recognize its technical merits.

Major point

The paper reads like a pharmacological dosing regime that sets up behavioural plasticity. Give this better detailing or clarity about the experiment is needed.

The methods imply that OP50 dosed with nicotine is used to set up the behavioural paradigm. My reading would be that 100 microliters of a stock is added to a 3 ml agar plate. This suggests 30mM nicotine are added to the bacteria. If this is the case this would be a dose that is strongly antimicrobial. An alternative interpretation of the methods is that 1mM is added to the OP50 and after an ill-defined diffusion into the agar reaches approx. 30 microM dosing. As the authors introduction nicely points out nicotine does many things across the possible dose ranges.

Given the 3-hr incubation period it would be important that the authors make clear statements about when animals were evaluated relative to the on-food dosing. Nicotine is charged and would not wash out quickly once accumulated in the worm. Do the authors think they are assaying in the absence on intra organismal nicotine. If this is not the case, I would be cautious about invoking interoceptive effects of nicotine when you are simply studying the impact of circulating levels of intoxicating nicotine.

The study has some interesting controls, but it does not report on how sustained the nicotine induced intoxication is on defecation program. It is interesting that the onset of effects is fast 15 mins. I would also caution that this can put gustatory cueing as a potential explanation. Given the authors prowess in behavioural recording it would be interesting to know about the nature of the bacterial lawns laced with nicotine, the behaviour of the worms while on nicotine (body length would be an obvious measure) and whether there is aversion when place on the nicotine laced lawns.

The description of the behavioural paradigm is clear. Intoxication on food followed by an ill-defined experimental recovery on agar arenas in the absence of food.

1. If the nicotine compromises the food lawns, there will be a distinct food quality.

2. If the high doses of nicotine impacts feeding behaviour. It will at the higher dose range: there will be a food intake difference.

3. If either of these pertain the interoceptive cueing (food levels) will contribute to or overlap with confounding nicotine effects.

In the experiments it would help if the authors were clear about whether DHBE is present during the behavioural intoxication or present when the cellular (imaging or electrophysiology assays are being performed. Again, the time between removal from intoxication 3 hrs and experimental measurement would always be informative.

In essence setting up the paradigm would help the reader better ascertain if the nicotine has driven a behavioural state or the behavioural state is being measured in the presence of pharmacologically relevant drug.

Minor points.

Was eat-2 tested in the screen for molecular determinants. It would be a useful check for some of the issues raised above.

Ric-7 would have been an interesting/useful addition to this data.

I agree that interpreting the data of the body wall muscle mutants is challenging but I think a preconditioning with levamisole would be especially useful as the body wall muscle contraction that nicotine might induce would be controlled for and add mechanistic clarity. This point is raised by the authors nice data highlighting a mechanosensitive cueing to the behavioural response.

useful ref

Invert Neurosci.

2018 Nov 7;18(4):14. doi: 10.1007/s10158-018-0219-1.

---

## [Decision Letter · Decision Letter 2]

9 Sep 2025

Dear Dr Gao,

Thank you for your patience while we considered your revised manuscript "Nicotine induces abnormal motor coupling through sensitization of mechanosensory circuit in C. elegans" for publication as a Research Article at PLOS Biology. This revised version of your manuscript has been evaluated by the PLOS Biology editors, the Academic Editor, and the original reviewers.

Based on the reviews, we are likely to accept this manuscript for publication, provided you satisfactorily address the remaining points raised by the reviewers. Please also make sure to address the following data and other policy-related requests.

IMPORTANT: Please ensure that your next revision incorporate the following additional changes:

--------------

**Title:

We suggest a slight tweak of your title, to the following: "Nicotine induces abnormal motor coupling through sensitization of a mechanosensory circuit in C. elegans"

**Data:

Thank you for providing source data through Zenodo. However, some items related to this still need to be addressed.

-- We noticed that the sheet labels in your Data Resource excel file do not seem to map exactly onto the supplementary figures; for example, there are two separate sheets labeled 'Fig1 S1' and 'Fig1 S2', which would seem related to 'S1 Fig.," but it is not clear exactly which figure panels go with each data column. Can you please re-label the sheets in your Data Resource file so that they correspond precisely to the Supplementary figures?

We expect to see numerical data for the following figure panels, most of which seem already to be present:

1D

2ABDFG

3CD

4BCE

5ABC

6BCEG

S1BCDEF

S2CEF

S3ABCDEF

S4ABCD

S5BD

S6BF

-- Please also cite the location of the data clearly in all relevant main and supplementary Figure legends, e.g. “The data underlying this Figure can be found in S1 Data” or “The data underlying this Figure can be found in https://doi.org/10.5281/zenodo.XXXXX”

-- Supplementary files (e.g., excel). Please ensure that all data files are uploaded as 'Supporting Information' and are invariably referred to (in the manuscript, figure legends, and the Description field when uploading your files) using the following format verbatim: S1 Data, S2 Data, etc. Multiple panels of a single or even several figures can be included as multiple sheets in one excel file that is saved using exactly the following convention: S1_Data.xlsx (using an underscore).

------------

We expect to receive your revised manuscript within two weeks.

*Published Peer Review History*

*Press*

Sincerely,

Taylor

Taylor Hart, PhD,

Associate Editor

thart@plos.org

PLOS Biology

Reviewer remarks:

Reviewer #1: I appreciate the new data and extensive text revisions. The revised manuscript is stronger, with several additions that enhance the study. For example, the authors added new controls on nicotine diffusion and food quality, included egl-8, aex-3, aex-2, and nlp-40 mutants to dissect the role of DMP phases, and provided a clearer explanation of methods such as how DMP-reversal coupling was measured, together with appropriate statistical analysis. These changes substantially improve the rigor of the work, even though the precise circuit connection between TRNs and AVA remains unresolved. With these improvements in place, I outline below a few remaining concerns.

1) ACR-16 regulation

In response to my prior comment regarding interpretation of the AVA reporter (Ptwk-40s::acr-16cDNA::gfp), the authors conclude that ACR-16 upregulation is mediated by a post-transcriptional mechanism (lines 513-523). However, this is difficult to reconcile with the data in Figures S5A and 5B. As described in lines 307-311, the transgenic strain expresses functional ACR-16 cDNA tagged with SL2d::GFP in AVA neurons. Because the SL2 trans-splicing element should produce GFP and ACR-16 as separate proteins, this construct would primarily reflect transcriptional changes rather than post-transcriptional regulation. It seems unlikely that GFP and ACR-16, expressed separately, would be subject to the same post-transcriptional control. Since ACR-16 upregulation is central to the proposed mechanism of nicotine sensitization of AVA activation and reversal behavior, this point requires clarification.

2) Discussion of "interoceptive effects" versus residual nicotine

While I appreciate the authors' clarification that they define "interoceptive effects" as internal sensing and neural processing of nicotine-related signals, I agree with Dr. O'Connor that a "circulating intoxicant" is not necessarily equivalent to a "sensed internal state." Whether lingering nicotine represents interoceptive sensing or simply residual intoxication remains only partially resolved. The current justification relies on a broad interpretation of interoception that may not align with stricter definitions. In other words, the mere presence of nicotine does not constitute an interoceptive process unless specific internal sensors are engaged, rather than passive pharmacological effects. That said, both interpretations are interesting. To avoid confusion for readers, I suggest the authors include a clearer discussion distinguishing between residual pharmacological effects and bona fide interoceptive processes.

3) Minor comments:

Line 140: Change "These results suggests…" to "These results suggest…".

Line 163: Change "This alignment allowed to precisely synchronize…" to "This alignment allowed us to precisely synchronize…".

Line 180: Change "nicotine act…" to "nicotine acts…".

Line 394: Change "To exam this…" to "To examine this…".

Line 423: Remove "of" and change to "nicotine enhances the signal strength from TRNs to AVA."

Line 429: Change "depending" to "depend".

Line 461: Change "exhibiting" to "exhibit".

Line 468: Change "This dues to…" to "This is due to…".

Line 501: Correct "fand" to "and".

Reviewer #2: I appreciate the thorough revision and responses.

My comments have been almost completely addressed. I have just two very minor follow-up comments:

Comment 2: For S1B, a t-test is not a good test for comparing proportions. Rather, a Z test, for example, would much more appropriate.

Comment 12: The extra data provided on the mechanosensory mutants and the clarification regarding e1611 is very informative. I recommend adding this to the manuscript.

Reviewer #3 [Vincent O'Connor]: I am more than impressed with the effort and quality embedded within the revised paper. This provides excellent reassurance and I hope important clarification and insight around dosing.

I leave it to the authors to decide if there paradigm could reflect an aversive conditioning.

I hope the authors can find time to assemble the knowledge around nicotine dosing for publication.

I learnt a lot from reading this paper and the ensuing efforts to address reviewers concerns.

---

## [Editor Report · Decision Letter 3]

17 Sep 2025

Dear Dr Gao,

Thank you for the submission of your revised Research Article "Nicotine induces abnormal motor coupling through sensitization of a mechanosensory circuit in C. elegans" for publication in PLOS Biology. On behalf of my colleagues and the Academic Editor, Piali Sengupta, I am pleased to say that we can in principle accept your manuscript for publication, provided you address any remaining formatting and reporting issues. These will be detailed in an email you should receive within 2-3 business days from our colleagues in the journal operations team; no action is required from you until then. Please note that we will not be able to formally accept your manuscript and schedule it for publication until you have completed any requested changes.

PRESS

Sincerely, 

Taylor Hart, PhD,

Associate Editor

PLOS Biology

thart@plos.org